# Polo-like kinase Cdc5 regulates Spc72 recruitment to spindle pole body in the methylotrophic yeast *Ogataea polymorpha*

Hiromi Maekawa[1,2]*, Annett Neuner[3], Diana Rüthnick[3], Elmar Schiebel[3], Gislene Pereira[4,5], Yoshinobu Kaneko[1]

[1]Graduate School of Engineering, Osaka University, Suita, Japan; [2]Faculty of Agriculture, Kyushu University, Fukuoka, Japan; [3]Zentrum für Molekulare Biologie der Universität Heidelberg, DKFZ-ZMBH Alliance, Heidelberg, Germany; [4]Centre for Organismal Studies, University of Heidelberg, Heidelberg, Germany; [5]Division of Centrosomes and Cilia, German Cancer Research Centre (DKFZ), DKFZ-ZMBH Alliance, Heidelberg, Germany

**Abstract** Cytoplasmic microtubules (cMT) control mitotic spindle positioning in many organisms, and are therefore pivotal for successful cell division. Despite its importance, the temporal control of cMT formation remains poorly understood. Here we show that unlike the best-studied yeast *Saccharomyces cerevisiae*, position of pre-anaphase nucleus is not strongly biased toward bud neck in *Ogataea polymorpha* and the regulation of spindle positioning becomes active only shortly before anaphase. This is likely due to the unstable property of cMTs compared to those in *S. cerevisiae*. Furthermore, we show that cMT nucleation/anchoring is restricted at the level of recruitment of the γ-tubulin complex receptor, Spc72, to spindle pole body (SPB), which is regulated by the polo-like kinase Cdc5. Additionally, electron microscopy revealed that the cytoplasmic side of SPB is structurally different between G1 and anaphase. Thus, polo-like kinase dependent recruitment of γ-tubulin receptor to SPBs determines the timing of spindle orientation in *O. polymorpha*.

DOI: https://doi.org/10.7554/eLife.24340.001

*For correspondence:
hmaekawa@agr.kyushu-u.ac.jp

**Competing interests:** The authors declare that no competing interests exist.

## Introduction

Segregation of sister chromatids into two daughter cells is pivotal to the proliferation of eukaryotic cells. Chromosome segregation is followed by cytokinesis, which results in physical separation of two daughter cells. In many organisms, the position of the mitotic spindle dictates the site of cytokinesis, which ensures the inheritance and maintenance of genomic information in the daughter cells. Astral microtubules or cytoplasmic microtubules (cMTs), which emanate from the spindle poles and extend to the cell cortex, have a principle role in positioning and orienting the spindle with respect to the polarity cues of the cell type. Mechanisms governing the spindle positioning/orientation have been studied in a number of systems. However, regulations that determine the timing of establishing the spindle orientation, or the position of the centrosome, the primary MT organizing centre (MTOC), in interphase, are not well understood (*Kiyomitsu, 2015*; *Woyke et al., 2002*).

Spindle positioning is of particular importance in the budding yeast S*accharomyces cerevisiae*, where the cleavage site is determined at the start of the cell cycle independently of the position of the mitotic spindle. Therefore, cells position the pre-anaphase spindle close to the bud neck and orient it along the mother-bud axis. As the spindle elongates in anaphase, one spindle pole

**eLife digest** Before a cell divides, it needs to duplicate its genetic material to provide the new daughter cell with a full set of genetic information. To do so, the cell forms a complex of proteins called the spindle apparatus, which is made up of string-like microtubules that divide the chromosomes evenly. In many organisms, the position of the spindle determines where in the cell this separation happens.

However, in baker's yeast, the location where the cell will divide is determined well before the spindle is formed. Unlike many other eukaryotic cells, these yeast cells divide asymmetrically and create buds that will form the new daughter cells. The position of this bud determines where the spindle should be located and where the chromosomes separate.

The spindle itself is then organised by a structure called the spindle pole body, which connects to microtubules inside the cell nucleus and microtubules in the cell plasma. Several proteins control where and how the spindle forms, including a protein called the spindle pole component 72, or Spc72 for short, and an enzyme called Cdc5. However, until now it was unclear how spindle formation is timed and controlled in other yeast species.

Now, Maekawa et al. have used fluorescent markers and time lapse microscopy to examine how the spindle forms in the yeast species *Ogataea polymorpha*, an important industrial yeast used to produce medicines and alcohol. The results show that in *O. polymorpha*, the positioning and orientation of the spindle only occurred very late in the cell cycle and the microtubules in the cell plasma remained unstable until the chromosomes were about to separate. This was linked to changes in the level of Spc72, which increased at the spindle pole body before the chromosomes separated and then dropped again. This was controlled by Cdc5.

Understanding when and where microtubules are formed is an important step in understanding how cells divide. This is the first example of a budding yeast that creates new microtubules in the cell plasma every time the cell divides. Unravelling the molecular differences between yeast species could lead to new ways to optimise the use of industrial yeasts like *O. polymorpha*, or to combat disease-causing ones.

DOI: https://doi.org/10.7554/eLife.24340.002

translocates into the bud to accomplish segregation of one set of chromosomes into the daughter cell (*Pereira and Yamashita, 2011*; *Markus et al., 2012*; *Winey and Bloom, 2012*).

In *S. cerevisiae*, the nuclear positioning and spindle orientation are regulated by two redundant pathways acting on cMTs, the Kar9 and dynein pathways (*Li et al., 1993*; *Miller and Rose, 1998*; *Winey and Bloom, 2012*). Concomitant deletions in components of both pathways result in lethality, whereas loss of one pathway can be compensated by the function of the other with moderate spindle orientation defects (*Miller and Rose, 1998*). Survival of single deletion mutants largely relies on the function of the spindle orientation checkpoint (SPOC) that retains cells in anaphase until the spindle orientation is corrected (*Bardin et al., 2000*; *Pereira et al., 2000*; *Caydasi and Pereira, 2012*).

Furthermore, MTs in *S. cerevisiae* are organized exclusively from the spindle pole body (SPB), which is the functional equivalent of animal centrosome. The SPB is a multilayered cylindrical organelle that is embedded in the nuclear envelope (NE) throughout the cell cycle (*Byers and Goetsch, 1974*; *Byers and Goetsch, 1975* )The outer plaque faces the cytoplasm and nucleates cMTs, whereas the inner plaque is inside the nucleus and organizes the nuclear MTs. The central plaque anchors and interconnects the outer and inner plaques (*O'Toole et al., 1999*; *Jaspersen and Winey, 2004*). In G1 phase, some fractions of the cMTs are organized from a modified region of the NE associated with one side of the SPB known as the half-bridge (*Byers and Goetsch, 1974*; *Byers and Goetsch, 1975*). Spc72, a γ-tubulin complex (γ-TuSC) receptor, is required for nucleating MTs at both the outer plaque and the half-bridge (*Chen et al., 1998*; *Knop and Schiebel, 1998*; *Wigge et al., 1998*; *Souès and Adams, 1998*). Localisation of Spc72 at the outer plaque is mediated by binding to Nud1, whereas Kar1 serves as a G1 specific binding site of Spc72 at the half-bridge (*Pereira et al., 1999*; *Gruneberg et al., 2000*). Spc72 also has a structural role as an integral part of the outer layer and as such localisation of Spc72 to the SPB and the ability to nucleate cMTs persist through the entire cell cycle (*Shaw et al., 1997*; *Pereira et al., 1999*; *Kosco et al., 2001*).

Importantly, Spc72, and hence cMTs, is not recruited for the formation of the SPB. New SPB acquires Spc72 and cMTs after the formation of a 1 μm long spindle (*Shaw et al., 1997*; *Segal et al., 2000*; *Juanes et al., 2013*).

In addition to the γ-tubulin complexes, Spc72 exerts a role in recruiting several other proteins to SPBs including Stu2, a microtubule-associated protein (MAP) of the XMAP215/Dis1 family, the SPOC kinase Kin4, as well as polo-like kinase Cdc5 (*Chen et al., 1998*; *Usui et al., 2003*; *Maekawa et al., 2007*; *Snead et al., 2007*). Cdc5 regulates multiple cellular functions including SPB duplication, progression through G2/M phase, promoting mitotic exit, and cytokinesis (*Shirayama et al., 1998*; *Hu et al., 2001*; *Song and Lee, 2001*; *Archambault and Glover, 2009*; *Elserafy et al., 2014*). Cdc5 is also involved in the regulation of spindle orientation in pre-anaphase and migration of the anaphase spindle (*Snead et al., 2007*; *Park et al., 2008*). Although Spc72 becomes highly phosphorylated during mitosis in a Cdc5-dependent manner, it is unclear whether this phosphorylation has a regulatory effect on Spc72 and/or cMTs (*Maekawa et al., 2007*; *Snead et al., 2007*).

The molecular mechanisms that control spindle orientation in *S. cerevisiae* have been well established. However, other species that employ the budding mode of cell division may have adopted different strategies. In the pathogenic yeast *Candida albicans,* the nucleus is located away from the bud neck in pre-anaphase cells (*Martin et al., 2004*; *Finley et al., 2008*). *C. albicans* and probably some of other species in Saccharomycotena may therefore have different mechanisms and regulations in this fundamental biological process.

*Ogataea polymorpha* (previously *Hansenula polymorpha*) is extensively used in industrial biotechnology, for the production of various pharmaceuticals in particular for its advantageous characteristics including methylotrophy, nitrate assimilation, availability of strong promoters, and low amount of secreted proteins (*Gellissen et al., 2005*; *Stöckmann et al., 2009* ). Another attractive property of *O. polymorpha* is its thermotolerant nature (up to approximately 50°C), which may reduce the cost of cooling in, for instance, bioethanol production that requires the treatment of raw materials at high temperature prior to fermentation. However, despite its importance, cell biology research on this organism remains limited. A better understanding of the molecular physiology of *O. polymorpha* is beneficial towards improving the abilities and characteristics of this yeast for a wide variety of applications.

Here, we describe cMT organization and its regulation during the cell cycle of the methylotrophic yeast *O. polymorpha*. Unlike *S. cerevisiae*, the pre-anaphase spindle is not readily positioned and oriented in *O. polymorpha* owing to the poorly organized cMTs at early cell cycle stages. The bottleneck of cMT nucleation/anchoring at SPBs occurs at the level of Spc72 recruitment to the SPBs, for which the polo-like kinase Cdc5 plays a crucial role. Consistent with the cell cycle dependent activity of cMTs, SPB structure also undergoes cell cycle dependent modification. Thus, our study shed light on the divergent nature of the temporal control of the cMT formation in yeast species.

## Results

### Nuclear positioning in *O. polymorpha* differs from that in *S. cerevisiae* and other budding yeast species

The nucleus is positioned close to the bud neck in large budded pre-anaphase cells of *S. cerevisiae* (*Figure 1A*). Similar organization was observed in other budding yeast species including *Candida glabrata*, *Kluyveromyces lactis*, *Pichia pastoris*, and *Yarrowia lipolytica* (*Figure 1—figure supplement 1*). Notably in *O. polymorpha*, however, nuclear position was not biased to the bud neck, although it remained in the mother cell body (*Figure 1A*, *Figure 1—figure supplement 1*). The phenotype resembled, but was more exaggerated than, that in *C. albicans* where the nucleus is located with a distance from the bud neck in pre-anaphase cells (*Martin et al., 2004*; *Finley et al., 2008*). A similar phenotype was observed in species closely related to *O. polymorpha* or *C. albicans* (*Figure 1—figure supplement 1*). Close examination of *O. polymorpha* revealed that the nucleus was located in the cell centre in 76.7% of G2/M cells whereas in the remainder of the cells it was off-centred with no bias towards the bud neck (*Figure 1A*, *Figure 1—figure supplement 2*). These results suggest that the nuclear position is not determined before anaphase onset in *O. polymorpha*. As a consequence, the early stages of anaphase occurred in the mother cell body (*Figure 1B*). This was further confirmed in cells expressing the α-tubulin gene (Tub1)-GFP and histone H3 gene (Hht1)-mRFP

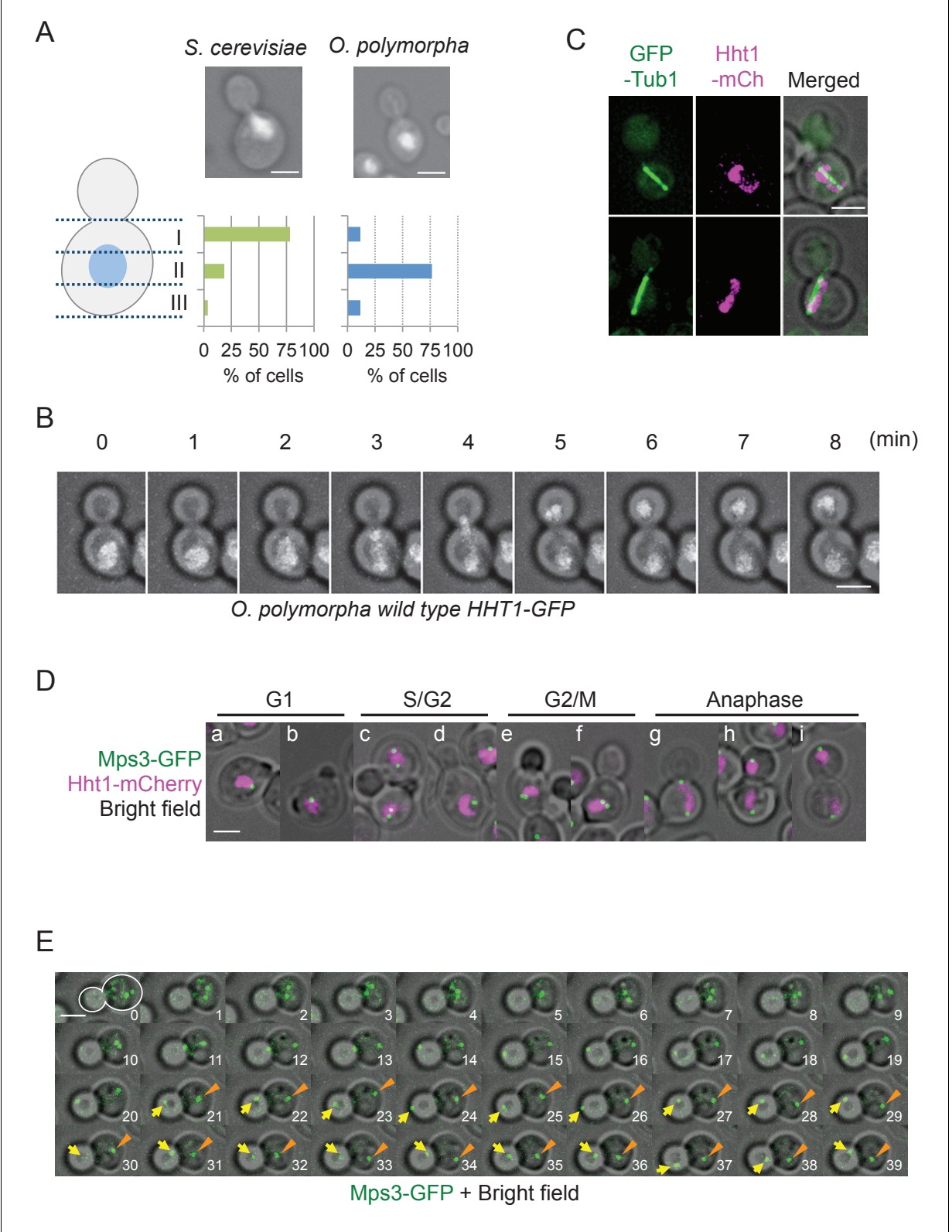

**Figure 1.** Nuclear positioning in *O. polymorpha*. (**A**) Nucleus is positioned in the cell centre in pre-anaphase cells of *O. polymorpha*. *S. cerevisiae* strain YPH499 and *O. polymorpha* type strain CBS4732 were grown in YPDS at 30°C. DNA was stained with DAPI. The positions of nuclei were as outlined in the cartoon shown on the left side of the subfigure. Scale bar, 2 µm. N = 60 (YPH499), 55 (CBS4732). Result of a similar experiment using *HHT1-GFP* cells is shown in *Figure 1—figure supplement 2*. (**B**) Time-lapse microscopy of histone H3 gene (*HHT1*)-GFP cells (HPH31). Anaphase onset judged by

*Figure 1 continued on next page*

*Figure 1 continued*

Hht1-GFP was observed at the 3 min timepoint. Shown are a merged figure of bright field images and deconvolved and projected GFP images. Scale bar, 2 μm. (C) Early anaphase cells (HPH164) with a single DNA mass along the spindle grown in YPDS at 30°C. Microtubules and DNA are visualized by GFP-Tub1 and Hht1-mCherry fluorescence, respectively. Scale bar, 2 μm. (D) Early anaphase cells (HPH1678) grown in YPDS at 30°C. SPB and DNA are visualized by Mps3-GFP and Hht1-mCherry fluorescence, respectively. Scale bar, 2 μm. (E) Time lapse microscopy of cells expressing SPB-GFP maker. *MPS3-GFP* cells (HPH1681) were grown in SD complete medium at 30°C. Consecutive sections were taken every 60 s. Shown are representative images of cells with an inappropriate angled spindle against the polarity axis. Yellow arrows and orange arrowheads point SPB. Shown are deconvolved and projected GFP images merged with bright field image. Scale bar, 2 μm. Another example is shown in *Figure 1—figure supplement 6*.
DOI: https://doi.org/10.7554/eLife.24340.003

The following figure supplements are available for figure 1:

**Figure supplement 1.** Nuclear position in different yeasts and their phylogenetic relationship.
DOI: https://doi.org/10.7554/eLife.24340.004
**Figure supplement 2.** Nuclear positioning in *O. polymorpha*.
DOI: https://doi.org/10.7554/eLife.24340.005
**Figure supplement 3.** Nuclear segregation in *O.polymorpha*.
DOI: https://doi.org/10.7554/eLife.24340.006
**Figure supplement 4.** SPB duplication initiates at the timing of bud emergence in *O. polymorpha*.
DOI: https://doi.org/10.7554/eLife.24340.007
**Figure supplement 5.** Time lapse microscopy of cells expressing SPB-GFP maker.
DOI: https://doi.org/10.7554/eLife.24340.008
**Figure supplement 6.** Behaviour of SPB during spindle alignment and mitosis.
DOI: https://doi.org/10.7554/eLife.24340.009

(monomeric red fluorescent protein) to visualize MTs and chromosomal DNA, respectively. Majority of the early anaphase spindles as judged according to their length (<5 μm) as well as a stretched DNA mass were located entirely in the mother cell body (97.4%, *Figure 1C*, *Figure 1—figure supplement 3A*). However, 94.7% of the early anaphase spindles were aligned along the bud-mother axis and almost all of the late anaphase spindles with two segregated DNAs were inserted into the bud (*Figure 1—figure supplement 3*), suggesting immediate and efficient orientation of the spindle during anaphase.

SPB position during the cell cycle was examined to clarify the spindle cycle relative to bud size in cells expressing the SPB marker (Mps3-GFP) (*Figure 1D*). Two SPB signals appeared in some of small/medium budded cells (*Figure 1D*, panel c, lower cell), suggesting that SPBs were duplicated at the timing of bud emergence or later, which is similar to that in *S. cerevisiae* although the precise cell cycle stage should be carefully determined. In the rest of small budded cells, one SPB signal was evident until large budded cells (*Figure 1D*, panel d). This may be because duplicated SPBs remained in a close proximity and could not be resolved by our fluorescence microscopy. Consistent with this, intensity of Mps3-GFP in medium/large-budded cells with a single SPB was much higher than that in unbudded G1 cells or cells with separated SPBs (*Figure 1—figure supplement 4*). Moreover, SPB in G1 cells as well as small budded cells was not in the defined position within the mother cell body (*Figure 1D*). Subsequent time lapse analysis revealed that after spindle assembly, ~1 μm long spindles remained at their central positions and loosely oriented toward the bud neck until shortly before anaphase onset (defined by the rapidly increase of pole-to-pole distance) (*Figure 1—figure supplement 5*). Anaphase initiated in the mother cell body (*Figure 1E*, 11 min, *Figure 1—figure supplement 6*, 3–4 min). These observation defined cells with 2SPBs in a < 2 μm distance as pre-anaphase cells. Spindle alignment was corrected around the time of (or shortly after) spindle elongation, followed by SPB insertion into the bud. After spindle breakdown, the SPB moved vigorously with no relationship to the polarity axis (*Figure 1E*, *Figure 1—figure supplement 6*).

## *O. polymorpha* cells contain only fewer cMTs

Lack of nuclear positioning and spindle orientation in pre-anaphase cells may indicate a very low number of cMTs at SPBs during this cell cycle window. To test this notion, we first investigated the MT organization during the cell cycle in *GFP-TUB1 HHT1-mCherry* cells (*Figure 2A and B*). Cell cycle stages were judged by the bud size and the number of DNA masses. cMTs were observed in all

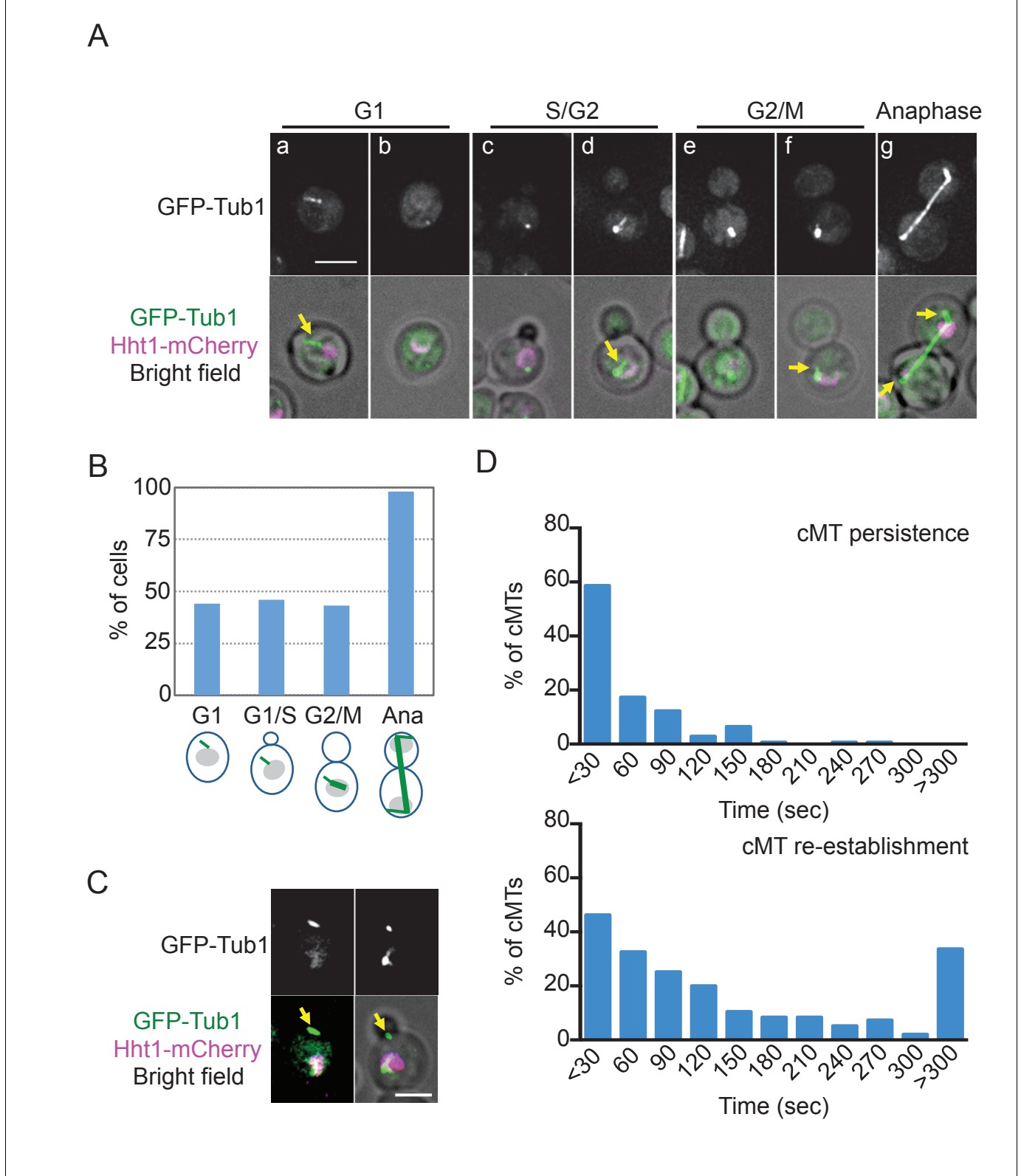

**Figure 2.** *O. polymorpha* cells contain fewer cytoplasmic microtubules (cMTs). (**A**) Wild-type cells (HPH164) were grown in YPDS medium at 30°C. Microtubules and DNA are visualized by GFP-Tub1 and Hht1-mCherry fluorescence, respectively. cMTs are marked by yellow arrows. a, b: unbudded G1 cell; c, d: preanaphase cell with monopolar nuclear MTs; e, f: preanaphase cell with bipolar spindle; g: anaphase cell. Scale bar, 2 µm. (**B**) Quantification of (**A**). G1, S/G2, G2/M, and anaphase represent unbudded cells, small budded cells with single unduplicated SPB, medium budded cells

*Figure 2 continued on next page*

*Figure 2 continued*

with 2 SPB, and large budded cells with an SPB in both the mother and the bud, and large budded cells with elongated spindle, respectively. n > 50 cells for each category. Ana, anaphase. (C) Cells containing the cMT detached from the SPB in (A). Scale bar, 2 μm. (D) Analysis of the duration of cMT persistence and cMT re-establishment time at SPBs in preanaphase cells with a bipolar spindle by time-lapse microscopy. *GFP-TUB1* cells (HPH194) were grown in SD complete medium at 30℃. Consecutive sections were taken every 30 s. Duration of continuous cMT presence was scored as cMT persistence, while the time between the loss of cMT and the acquisition of a new cMT was scored as cMT re-establishment. Total recording time was 38460 s.

DOI: https://doi.org/10.7554/eLife.24340.010

The following figure supplements are available for figure 2:

**Figure supplement 1.** Time-lapse microscopy of *GFP-TUB1 HHT1-mCherry* cells.

DOI: https://doi.org/10.7554/eLife.24340.011

**Figure supplement 2.** Time-lapse microscopy of *GFP-TUB1* cells.

DOI: https://doi.org/10.7554/eLife.24340.012

**Figure supplement 3.** Time-lapse microscopy of *GFP-TUB1* cells.

DOI: https://doi.org/10.7554/eLife.24340.013

anaphase cells, whereas less than 50% of G1 and pre-anaphase cells carried cMTs. Furthermore, cMTs that apparently did not associate with the SPB were observed in 13.8% of cells prior to anaphase (*Figure 2C*). Time lapse analysis revealed that detached cMTs remained in the cytoplasm for only a short period of time before depolymerized (*Figure 2—figure supplement 1*). This situation is in stark contrast to that in *S. cerevisiae* where almost all cells exhibit cMTs that are stably associated with the SPB during the cell cycle (*Shaw et al., 1997*; *Kosco et al., 2001*).

The small number of observed cMTs might have arisen because of reduced cMT nucleation. Another and not mutually exclusive possibility considers that cMTs might not be stably anchored to the SPB and thus might not persist over long periods. To test these possibilities, we performed time lapse experiments with cells expressing *GFP-TUB1*, in which cMTs were observed during 20.6% of the recorded time points although the majority (>80%) did not persist longer than 30 s (*Figure 2D*, cMT persistence, *Figure 2—figure supplement 2*). These results suggest that cMTs are short-lived during early stages of the cell cycle. Once cMTs were lost, a relatively long time was required until new MTs appeared at the SPB (*Figure 2D*, cMT re-establishment; median value 90.0 s, average 173.8 ± 192.4 s). Thus, cMTs are less frequently nucleated and unstable at early stages of the cell cycle. Acquired cMTs efficiently corrected the spindle orientation in pre-anaphase cells, suggesting that the spindle orientation is regulated largely at the level of cMT acquisition (*Figure 2—figure supplement 3*).

## Organization of the SPB structure on the cytoplasmic side is cell cycle dependent

Next, we examined the SPB structure in G1 and anaphase by electron microscopy. An electron dense SPB-like structure was evident in all cells examined, while a half-bridge-like structure—which plays an important role in cMT organization in G1 of *S. cerevisiae*—was not clearly observed (*Figure 3*). Anaphase SPBs had an additional thin layer in the cytoplasm that resembled the outer plaque of *S. cerevisiae* SPB (*Figure 3A and B*) (*Byers and Goetsch, 1974*; *Byers and Goetsch, 1975*). In contrast, there was no detectable outer plaque in G1 SPBs (*Figure 3C and D*). These results suggested that *O. polymorpha* SPBs undergo structural cycling in every cell cycle.

Our attempt to arrest cells in late G1 by introducing *cdc28-as* allele, which arrest *S. cerevisiae* cells in late G1 with a single SPB and satellite, was failed probably because of insufficient inhibition of the kinase (*Figure 3—figure supplement 1*). However, inhibitor addition delayed cell cycle progression leading to the accumulation of cells with unseparated SPBs. This allowed us to examine the structure of side-by-side SPBs by EM (*Figure 3—figure supplement 2*). All six side-by-side SPBs had outer plaques which were similar to that in nocodazole arrested cells, albeit some of those were somewhat fuzzy. The result suggested that the SPB structure on cytoplasmic side is reconstructed before spindle formation. An additional electron dense cloud was observed on the cytoplasmic side of nuclear envelope between two SPB bodies, which resembled the half-bridge/bridge structure of *S. cerevisiae* SPB (*Figure 3—figure supplement 2A, B and E*; orange arrowheads). It was not clear whether this structure was present at other cell cycle stages. A better synchronization method is

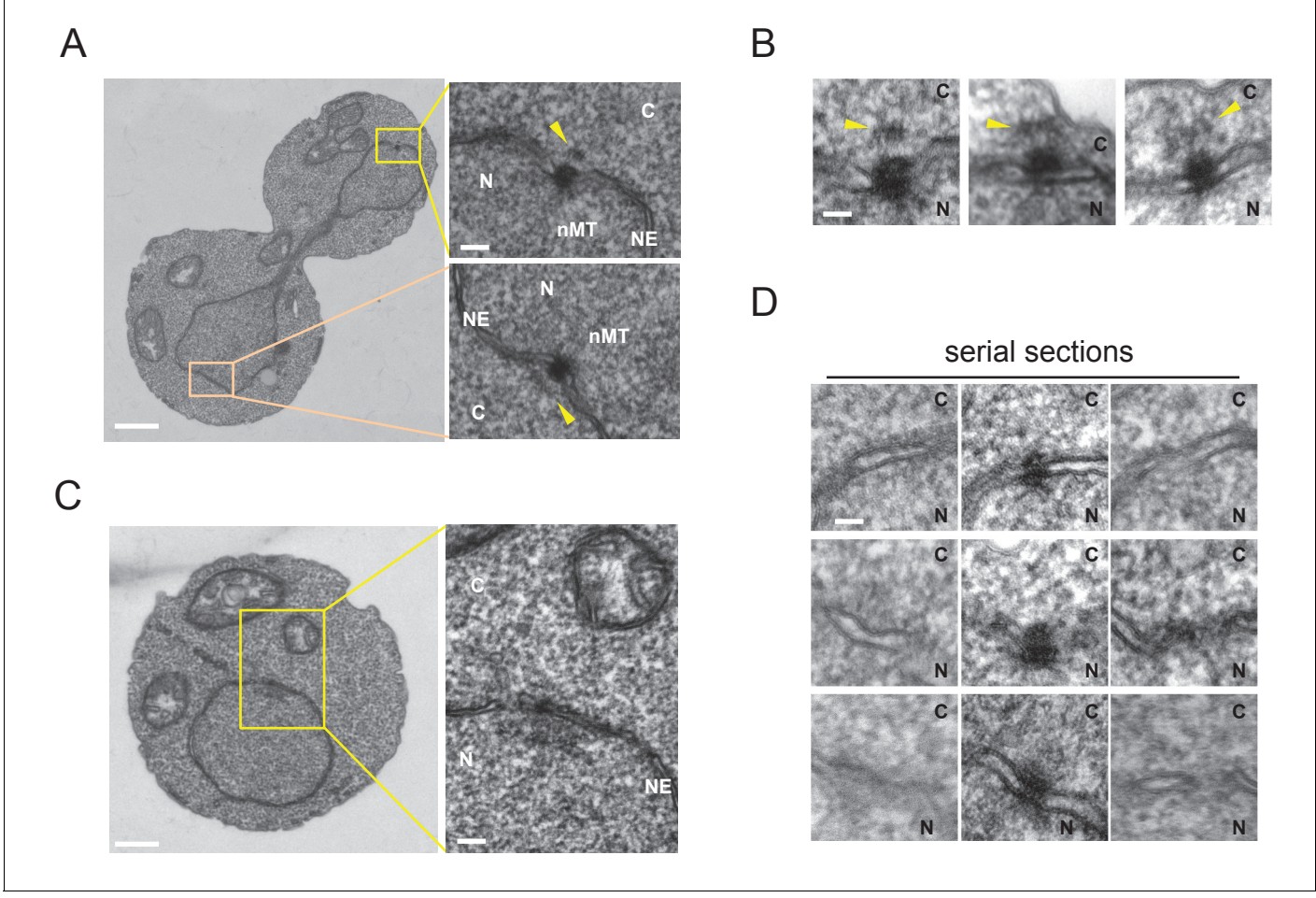

**Figure 3.** Cytoplasmic structure of SPB is regulated during the cell cycle. Electron microscopy (EM) of thin serial sections of cells in G1 and anaphase. Wild-type cells (BY4329) were grown to log phase at 30°C in YPDS and then prepared for EM. Indicated are the cytoplasm (C), nucleus (N), nuclear envelope (NE), and nuclear microtubules (nMT). (**A, B**) Representative SPBs in anaphase cells (n = 10). The SPB in the mother does not appear in the section shown as the main image. The pale orange rectangle in the mother merely indicates the position of the lower inset which is the image of the section containing the SPB in the mother. (**C, D**) representative SPBs in unbudded G1 cells (n = 10). Consecutive three sections are shown in (**D**). Scale bars of the main images in (**A**) and (**C**) represent 1 μm. Scale bars of the insets in (**A**) and (**C**), i.e., (**B**) and (**D**), represent 100 nm.

DOI: https://doi.org/10.7554/eLife.24340.014

The following figure supplements are available for figure 3:

**Figure supplement 1.** Phenotype of *O. polymorpha cdc28-as* cells.

DOI: https://doi.org/10.7554/eLife.24340.015

**Figure supplement 2.** Side-by-side SPBs contains outer plaque.

DOI: https://doi.org/10.7554/eLife.24340.016

required to determine the fine structure and the precise timing of emergence/disappearance of the outer plaque during the cell cycle.

## Spc72 associates with SPB in a cell cycle-dependent manner

Lack of the outer plaque in G1 prompted us to search for SPB components whose association with SPBs was cell cycle dependent. In *S. cerevisiae*, core SPB components are found at SPBs throughout the mitotic cell cycle including central plaque components (Spc42, Spc29), outer plaque components (Cnm67, Nud1), half-bridge components (Sfi1, Cdc31, Mps3, and Kar1), and membrane anchors (Ndc1, Nbp1, Mps2, and Bbp1) (*Winey et al., 1991*; *Spang et al., 1995*; *Bullitt et al., 1997*; *Brachat et al., 1998*; *Wigge et al., 1998*; *Chial et al., 1998*; *Adams and Kilmartin, 1999*; *Elliott et al., 1999*; *Schramm et al., 2000*; *Kilmartin, 2003*; *Jaspersen et al., 2002*; *Araki et al.,*

*2006*). The γ-TuSC recruiting factors Spc110 and Spc72 also represent core components of SPB in the inner and outer plaques, respectively (*Knop and Schiebel, 1997*; *Knop and Schiebel, 1998*). BLAST and HMMER searches have identified putative orthologues of genes for Mps3, Sfi1, Spc72, Spc110, and Nud1 as well as γ-TuSC components, Tub4, Spc97, and Spc98 in the *O. polymorpha* genome (*Altschul et al., 1990*; *Sobel and Snyder, 1995*; *Geissler et al., 1996*; *Knop and Schiebel, 1997*; *Eddy, 1998*; *Maekawa and Kaneko, 2014*; *Riley et al., 2016*). SPB-like localisation was verified by expressing GFP or mRFP-fused version of these proteins. GFP or mRFP dot-like signals of Tub4, Spc98, Mps3, Sfi1, and Nud1 were observed in most of the cells, suggesting that they represent constitutive components of the SPB throughout the cell cycle (*Figure 4A*, *Figure 4—figure supplement 1*). In contrast, the Spc72 signal was either weak or absent in cells at early cell cycle stages, whereas all anaphase cells carried two strong SPB signals (*Figure 4A*, *Figure 4—figure supplement 1*). Deletion of *SPC72* in *S. cerevisiae* results in severe growth defects or lethality depending on the strain background. To evaluate the effect of *SPC72* deletion in *O. polymorpha*, *SPC72/spc72Δ:: natNT2* heterozygous diploid cells were subjected to tetrad dissection analysis. Notably, 21 out of 29 tetrads yielded one or two viable colonies, all of which were sensitive to nourseothricin (*Figure 4—figure supplement 2*). Microscopic inspection revealed that 91.7% of spores with *spc72Δ:: natNT2* genotype derived from tetrads that gave two viable nourseothricin-sensitive colonies were germinated. These results suggested that *SPC72* is essential for growth in *O. polymorpha*.

To more precisely evaluate the amount of Spc72 at the SPB, images were obtained in logarithmically growing wild-type cells carrying *SPC72-GFP MPS3-mRFP* and the GFP intensity at the SPB was quantified (*Figure 4B*). The GFP signal was 2.5 times weaker in cells with a short spindle than in anaphase cells (p<0.001). In contrast, Nud1, which comprises the putative binding site of Spc72 on the outer plaque as suggested by the direct interaction between orthologues of these proteins in *S. cerevisiae*, did not show this trend, nor did Sfi1, a half-bridge component (*Figure 4B*) (*Gruneberg et al., 2000*; *Kilmartin, 2003*). High intensity of Sfi1-GFP signal in S/G2 cells most likely arose from SPBs that were duplicated but not yet separated. These results suggest that Spc72 is cell cycle regulated and the incorporation of Spc72 into SPBs may be the key step to stabilize cMTs. To further confirm this notion, time-lapse microscopy was carried out to determine the timing of Spc72 association with SPBs (*Figure 4C*, *Figure 4—figure supplement 3*). In all cells that progressed into anaphase, an Spc72-GFP signal became detectable <4 min prior to the initiation of anaphase (average 3.68 ± 1.74 min, n = 14) (*Figure 4C*, orange arrowheads). Within 5 min after appearance of the Spc72-GFP signal, spindle orientation was corrected when it had not done already (*Figure 4—figure supplement 3*, average 3.50 ± 1.61 min, n = 12); therefore, one half part of an anaphase nucleus was successfully inserted into the bud. Thus, Spc72 accumulates at SPB in early mitosis, most likely in metaphase, and remains high during anaphase. As cells exit from mitosis and entre the next cell cycle, Spc72-GFP signal was gradually decreased at SPBs with the timing that varied from cell to cell. This difference of timing may explain the relatively high and variable intensity of Spc72-GFP at SPB in G1 cells (*Figure 4B*). However, in all cases, Spc72-GFP levels reached a minimum well before short spindle was formed (*Figure 4*, *Figure 4—figure supplement 4*).

If low abundance of Spc72 at the SPB is the reason underlying cMT instability, higher expression of Spc72 might increase the level of Spc72 at the SPB and consequently raise the number of cMTs, and thereby promote positioning the nucleus close to the bud neck at early stages of the cell cycle. We expressed the *SPC72-GFP* gene from a strong constitutive *TEF1* promoter in cells whose endogenous *SPC72* was also fused to *GFP* and examined the position of the SPB relative to the bud neck in pre-anaphase cells as a readout of cMT function (*Kiel et al., 2007*). Overexpressed Spc72-GFP was efficiently targeted to SPB because a strong Spc72-GFP signal was observed in cells carrying the P$_{TEF1}$-*SPC72-GFP* gene but not in wild-type cells during G1 and G2/M phases (*Figure 5A*, *Figure 5—figure supplement 1*). In cells overexpressing *SPC72-GFP*, SPB was positioned close to the bud neck, which is reminiscent of the SPB position in *S. cerevisiae* (*Figures 1A* and *5B*), and cMTs were more often observed (*Figure 5C and D*). Together, these results strongly supported our hypothesis that cMT organization is regulated at the level of repeated Spc72 recruitment to the SPB in every cell cycle.

cMT play important roles in yeast mating and karyogamy, which are initiated in G1. Because mating is triggered by nutrient starvation in *O. polymorpha*, we examined cMTs and Spc72 in nutrient starved cells. Interestingly, while Spc72 was accumulated at SPBs, cMTs were not observed

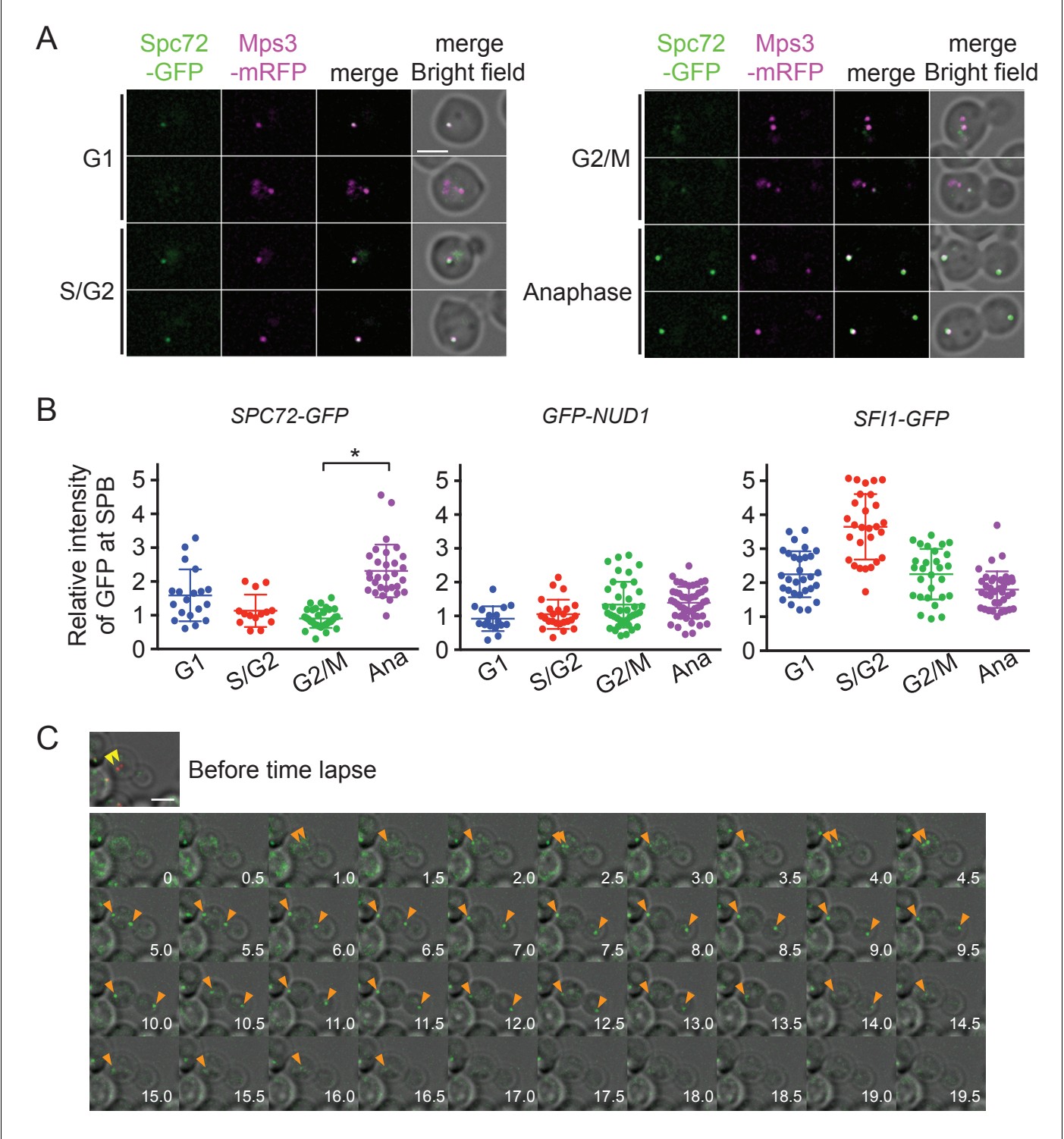

**Figure 4.** Accumulation of Spc72 at SPBs is cell cycle dependent. (A) Cell cycle dependent localization of Spc72-GFP. *SPC72-GFP MPS3-mRFP* cells (HPH1394) were grown in SD complete medium at 30°C. Cell cycle stages are as shown in *Figure 2B*. Mps3-mRFP is a marker for SPB. Scale bar, 2 μm. (B) Quantification of the Spc72/Nud1/Sfi1 SPB signal of cells at different cell cycle stages. Yeast strains HPH972, HPH1396, and HPH1400 were used. Signal intensities were background-subtracted. Statistical significance of the difference between 2 SPBs and anaphase was determined by the student t-test and is indicated by an asterisk. Error bars indicate SD. n = 95, 127, and 124 for Spc72, Nud1, and Sfi1, respectively. Note that intensity of some of Sc72-GFP signals in G1 was high, because it decreased only gradually at SPB during the end of mitosis and the following G1 as shown in (C). (C) Time-lapse microscopy of *SPC72-GFP MPS3-mRFP* cells (HPH1394). Images were taken every 30 s. RFP signal was captured only before staring the time-lapse

*Figure 4 continued on next page*

*Figure 4 continued*

series. Anaphase onset judged by sudden spindle elongation was observed at the 5 min timepoint. Spindle orientation was corrected between 4.5 min and 5 min timepoints. Yellow arrowheads indicate the position of Mps3-mRFP before the image capture. Orange arrowheads indicate Spc72-GFP signals at SPBs. Shown are deconvolved and projected images. Scale bar, 2 μm. Another example is presented in *Figure 4—figure supplement 1*.

DOI: https://doi.org/10.7554/eLife.24340.017

The following figure supplements are available for figure 4:

**Figure supplement 1.** SPB localization of *O. polymorpha* orthologues of SPB components and γ-TuSC components.

DOI: https://doi.org/10.7554/eLife.24340.018

**Figure supplement 2.** *SPC72* of *O. polymorpha* is essential for growth.

DOI: https://doi.org/10.7554/eLife.24340.019

**Figure supplement 3.** Time-lapse microscopy of SPC72-GFP cells.

DOI: https://doi.org/10.7554/eLife.24340.020

**Figure supplement 4.** Time-lapse microscopy of SPC72-GFP cells.

DOI: https://doi.org/10.7554/eLife.24340.021

**Figure supplement 5.** Stu2 binding domain of Spc72 is conserved among budding yeasts.

DOI: https://doi.org/10.7554/eLife.24340.022

(*Figure 5—figure supplement 2*). Thus, specific mechanism may regulate Spc72 and cMT organization under such conditions.

## SPB association of Spc72 is dependent on the polo-like kinase Cdc5

Spc72 might be regulated at the level of protein expression. To synchronize cells for monitoring changes in protein levels during the cell cycle, we transferred the recently developed auxin-inducible degradation (AID)-degron system to *O. polymorpha* (*Nishimura and Kanemaki, 2014*). *CDC5* encodes the only polo-like kinase in yeast. It is thus essential for growth in *S. cerevisiae*, and its inactivation causes cell cycle arrest in late anaphase (*Kitada et al., 1993*). Similarly, a single *CDC5* orthologue was identified in the *O. polymorpha* genome (*OpCDC5*). Logarithmically growing cells carrying a *3mAID*-tagged version of *CDC5* were arrested as large budded cells by incubation in the presence of auxin and then released into fresh medium without auxin to resume the cell cycle (*Figure 6A*). Spc72 protein abundance did not fluctuate as cells entered into anaphase and proceeded into the following cell cycle (*Figure 6A*, *Figure 6—figure supplement 1A*). Furthermore, the Spc72 band migrated slower in nocodazole-arrested cells than that in asynchronous cells (*Figure 6—figure supplement 1B*). These results suggested that either post-translational modification of Spc72 or regulation of Spc72 binding proteins might be utilized to achieve cell cycle dependency of SPB localisation.

Furthermore, we noticed that the GFP intensity of Spc72-GFP at the SPB was significantly lower in cells arrested by Cdc5-depletion than that in cells after re-accumulation of Cdc5 in both metaphase (SPB distance <2 μm, p<0.0001) and anaphase (SPB distance >4 μm, p<0.0001) (*Figure 6B and C*). Strong dependency of SPB binding of Spc72 on Cdc5 was further confirmed in metaphase-arrested cells by nocodazole (*Figure 6D*, *Figure 6—figure supplement 2*). These results suggested that the stable association of Spc72 requires Cdc5 kinase, the activity of which is likely cell cycle-regulated.

Spc72 is phosphorylated by Cdc5 kinase in *S. cerevisiae* (*Maekawa et al., 2007*; *Snead et al., 2007*), which prompted us to investigate whether Spc72 is subjected to a Cdc5-dependent phosphorylation in *O. polymorpha*. Our gel electrophoresis analyses of nocodazole-arrested cells suggested that Spc72 of *O. polymorpha* is subjected to post-translational modifications in a Cdc5-dependent manner because the Spc72 band was more smeared and migrated slightly slower in wild type and *CDC5*-overexpressing cells as compared to that in Cdc5-depleted cells (*Figure 6—figure supplement 3A*). Although this difference was largely lost during preparation of cell extract, the λ-phosphatase treatment revealed in vivo phosphorylation of Spc72 that was independent of Cdc5 (*Figure 6—figure supplement 3B*), suggesting that Cdc5 contributes to a subset of phosphorylations of Spc72. Thus, Spc72 is phosphorylated at multiple sites, only some of which depend on Cdc5.

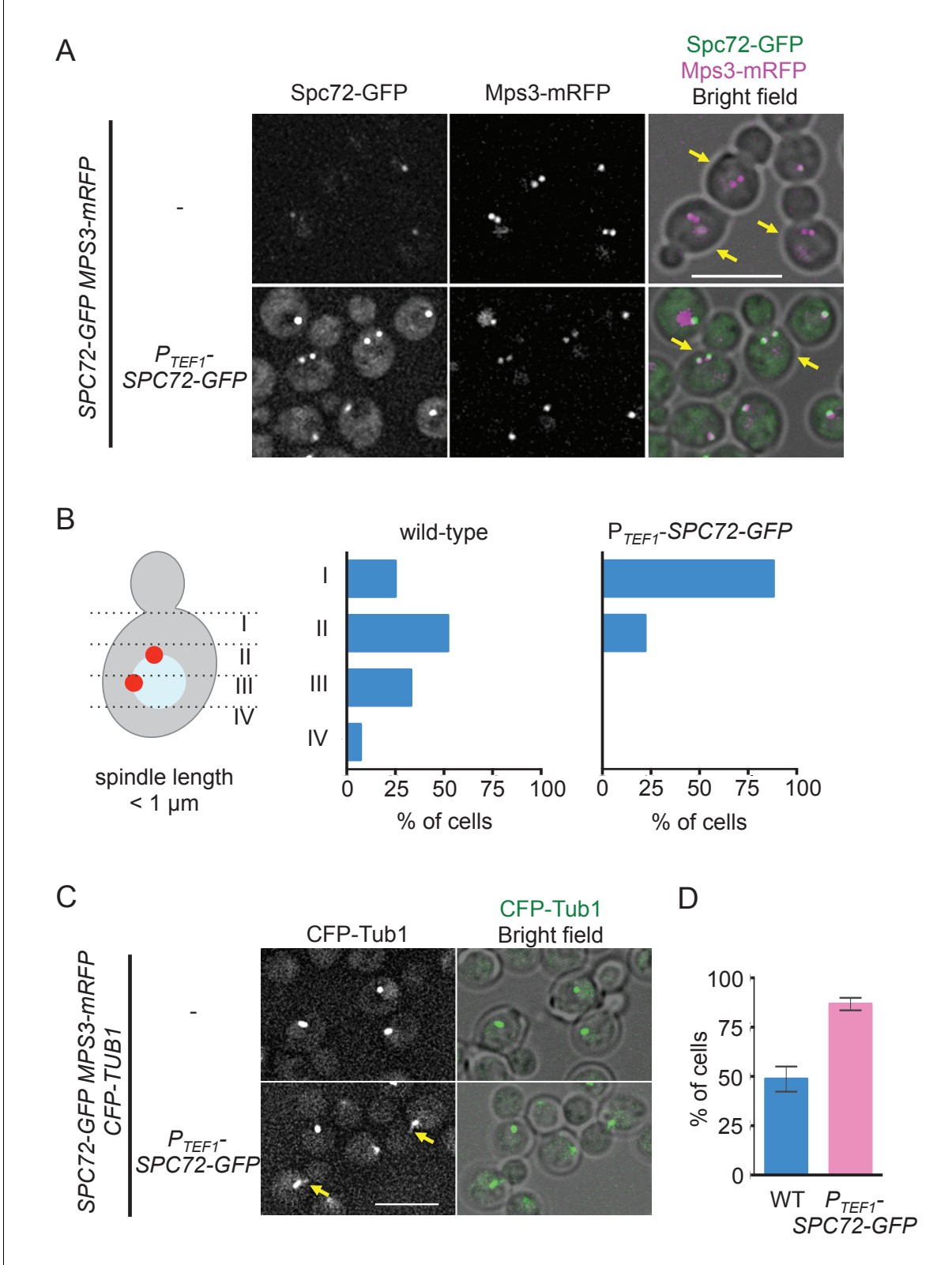

**Figure 5.** Overexpression of Spc72 converts the *O. polymorpha* type of nuclear position to the *S. cerevisiae* type. (**A**) Overexpressed Spc72-GFP were recruited to SPB at all stages of the cell cycle. *SPC72-GFP* was expressed from a strong promoter of the *TEF1* gene (HPH1393). Enrichment of Spc72 to SPB was evident in G1 cells and cells with short spindles (SPB distance <1 μm, yellow arrows) compared with images of wild-type cells (HPH1394). Mps3-mRFP marks SPB. Scale bar, 5 μm. (**B**) SPB is positioned close to the bud neck in G2/M cells carrying P_TEF1-*SPC72-GFP*. The position of the SPB

*Figure 5 continued on next page*

Figure 5 continued

closer to the bud was as outlined in the cartoon shown on the left side of the subfigure. Strains used were HPH1393 (n = 110) and HPH1394 (n = 117). (C) Overexpression of Spc72 stimulated cMT acquisition. MTs were visualized with *CFP-TUB1* in wild type and cells overexpressing *SPC72-GFP* (HPH1653 and HPH1652, respectively) were grown in YPDS medium at 30°C. Images were captured only for CFP and brightfield. Scale bar, 5 µm. (D) Quantification of (C). Presence/absence of cMTs was scored in cells with short spindle. Shown is the average of three independent experiments. Error bars indicate SD. n > 100. Average of three independent experiments.

DOI: https://doi.org/10.7554/eLife.24340.023

The following figure supplements are available for figure 5:

**Figure supplement 1.** Spc72-GFP binds to SPBs in G1 and G1/S cells when overexpressed.
DOI: https://doi.org/10.7554/eLife.24340.024

**Figure supplement 2.** Spc72-mRFP accumulate at SPB in starvation conditions.
DOI: https://doi.org/10.7554/eLife.24340.025

## Cdc5 localises to NE and SPB during mitosis

We next examined the localisation of Cdc5 during the cell cycle. Distinctive localization of Cdc5-GFP became apparent after S phase and was lost prior to or during the following G1 phase (*Figure 7A and B*, *Figure 7—figure supplement 1*). Nuclear and NE localisation appeared at early stages of the cell cycle and persisted until the end of mitosis. In addition, a fraction of the GFP signals appeared to overlap with those of SPBs during mitosis. Neither NE nor SPB-associated GFP signals were detected in unbudded or small budded cells. Notably, SPB signal may arise from the SPB itself, NE surrounding the SPB, or kinetochores that cluster close to the SPB during interphase and anaphase in yeast (*Jin et al., 1998*). Therefore, in order to clarify on which side of SPB the Cdc5-GFP signal resided, we employed structured illumination microscopy (SIM). Localisation was investigated in metaphase-arrested cells with nocodazole where Cdc5-GFP signal was observed at SPB as well as in nucleus (*Figure 7—figure supplements 1B* and *2*), and Spc72, and Spc110 were used as references for the cytoplasmic and nuclear side of SPB, respectively. Spc72-GFP and Spc110-tdTomato signals were clearly distinguished in 58% of the cells, which verified that our method could discriminate signals in the cytoplasmic and the nuclear side of SPB in >50% of cells (*Figure 7C*). The resolution of both signals probably depends on the orientation of the SPB (top versus side view). Only the SPB side view will resolve Spc72-GFP and Spc110-tdTomato signals at SPBs. Cdc5-GFP overlapped with Spc110-tdTomato in 53% of cells, which is similar to the degree of co-localisation observed between Spc72 and Spc110. In contrast, the Cdc5-GFP signal overlapped with Spc72-tdTomato in 87% of cells (*Figure 7D*). These results suggest that Cdc5-GFP locates at the position on SPBs closer to Spc72 than to Spc110, indicating that Cdc5-GFP signal arises from the cytoplasmic side of SPB. Thus, Cdc5 likely becomes first localised to the nucleus and the NE in G2, and then in mitosis to the cytoplasmic side of SPBs. The timing of Cdc5 binding to SPBs coincides well with the recruitment of Spc72 to SPBs.

## *CDC5* overexpression accelerates the Spc72 recruitment to SPB

To further confirm the significance of Cdc5 kinase in the recruitment of Spc72 to SPBs, we constitutively expressed the *CDC5* gene at high level. While Cdc5 expression showed no effect on the protein level of Spc72-GFP (*Figure 8—figure supplement 1*), Spc72-GFP intensity at the SPB was higher in metaphase-arrested cells following ectopic expression of *CDC5* from the *TEF1* promoter than in wild-type cells (p<0.0001) (*Figure 8A and B*). In the similar analysis performed in asynchronously growing cells, accumulation of Spc72-GFP at SPB was significantly higher at all stages of the cell cycle in cells overexpressing *CDC5* than in wild type cells (p<0.01 for G1 cells; p<0.0001 for S/G2, G2/M, and anaphase cells), with the strongest effect observed in G2/M phase (*Figure 8C and D*), and cMTs were more prevalent (*Figure 8E and F*). As a consequence, the SPB positioned closer to the bud neck (*Figure 8G*) and the spindle was at an angle within 30° with respect to the mother-bud axis in 83% of pre-anaphase cells overexpressing *CDC5* compared with 48% in wild type cells (*Figure 8—figure supplement 2*). These results suggest that the stable SPB association of Spc72 is restricted to the time period where Cdc5 kinase activity is sufficiently high.

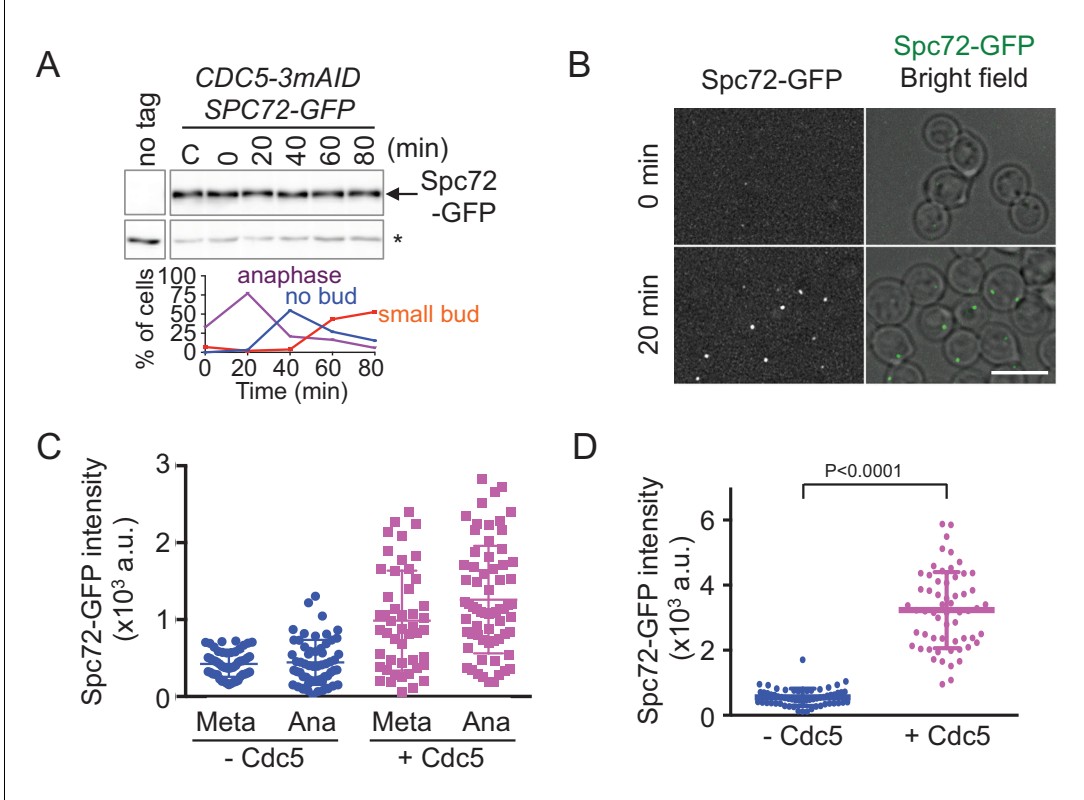

**Figure 6.** The recruitment of Spc72 to SPBs is dependent on the polo-like Cdc5 kinase. (A) Abundance of Spc72 does not fluctuate during the cell cycle. Logarithmically growing *SPC72-GFP CDC5-3mAID* P$_{CDC28}$-*OsTIR* cells (HPH1380) were synchronized with IAA followed by release. Samples were analysed by immunoblotting for Spc72-GFP. Comparable intensities of unspecific signal in immunoblotting (asterisk) indicate equal loading of samples. Budding index and mitotic index (DAPI) were determined over time. n > 100 cells per time point. (B) *SPC72-GFP CDC5-3mAID* P$_{CDC28}$-*OsTIR* cells (HPH1380) were synchronized and released as in (A). Images were captured without fixation. Shown are deconvolved and projected images. Time after release is indicated on the left. Scale bar, 5 μm. (C) Quantification of Spc72-GFP intensity at SPBs in (B). Signal intensities were background-subtracted. Statistical significance of the difference between 2 SPBs and anaphase was determined by the student t-test. – Cdc5: before the release;+Cdc5: 20 min after the release. Error bars indicate SD. n > 50 cells per time point. (D) Accumulation of Spc72 at SPBs in metaphase depends on Cdc5 function. *SPC72-GFP CDC5-3mAID* P$_{CDC28}$-*OsTIR* cells (HPH1380) were arrested with nocodazole in the presence (− Cdc5) or absence (+Cdc5) of IAA and Spc72-GFP signal at SPBs was quantified. Signal intensities were background-subtracted. Error bars indicate SD. n > 50 cells.

DOI: https://doi.org/10.7554/eLife.24340.026

The following figure supplements are available for figure 6:

**Figure supplement 1.** Spc72 protein level does not change at different cell cycle stages.
DOI: https://doi.org/10.7554/eLife.24340.027
**Figure supplement 2.** Localization of Spc72 to SPBs depends on Cdc5 activity.
DOI: https://doi.org/10.7554/eLife.24340.028
**Figure supplement 3.** Spc72 phosphorylation is partly dependent on Cdc5.
DOI: https://doi.org/10.7554/eLife.24340.029

## Discussion

The mode of cell division by budding represents a type of asymmetric cell division. The mechanism to achieve a high-fidelity of chromosome segregation in such a situation has been a focus of interest and has been investigated intensively in *S. cerevisiae*. These process was recently studied in the other ascomycetous yeast *C. albicans* as well as in the yeasts *Cryptococcus neoformans* and *Ustilago maydis* in another phylum of fungi, Basidiomycota (*McCully and Robinow, 1972a*; *McCully and Robinow, 1972b*; *Kopecká et al., 2001*; *Steinberg et al., 2001*; *Woyke et al., 2002*; *Straube et al., 2003*; *Martin et al., 2004*; *Finley et al., 2008*). Although the movement of the nucleus during the cell cycle differs between ascomycetous and basidiomycetous yeasts, it is commonly positioned close to the bud neck in both phyla prior to chromosome segregation. We report

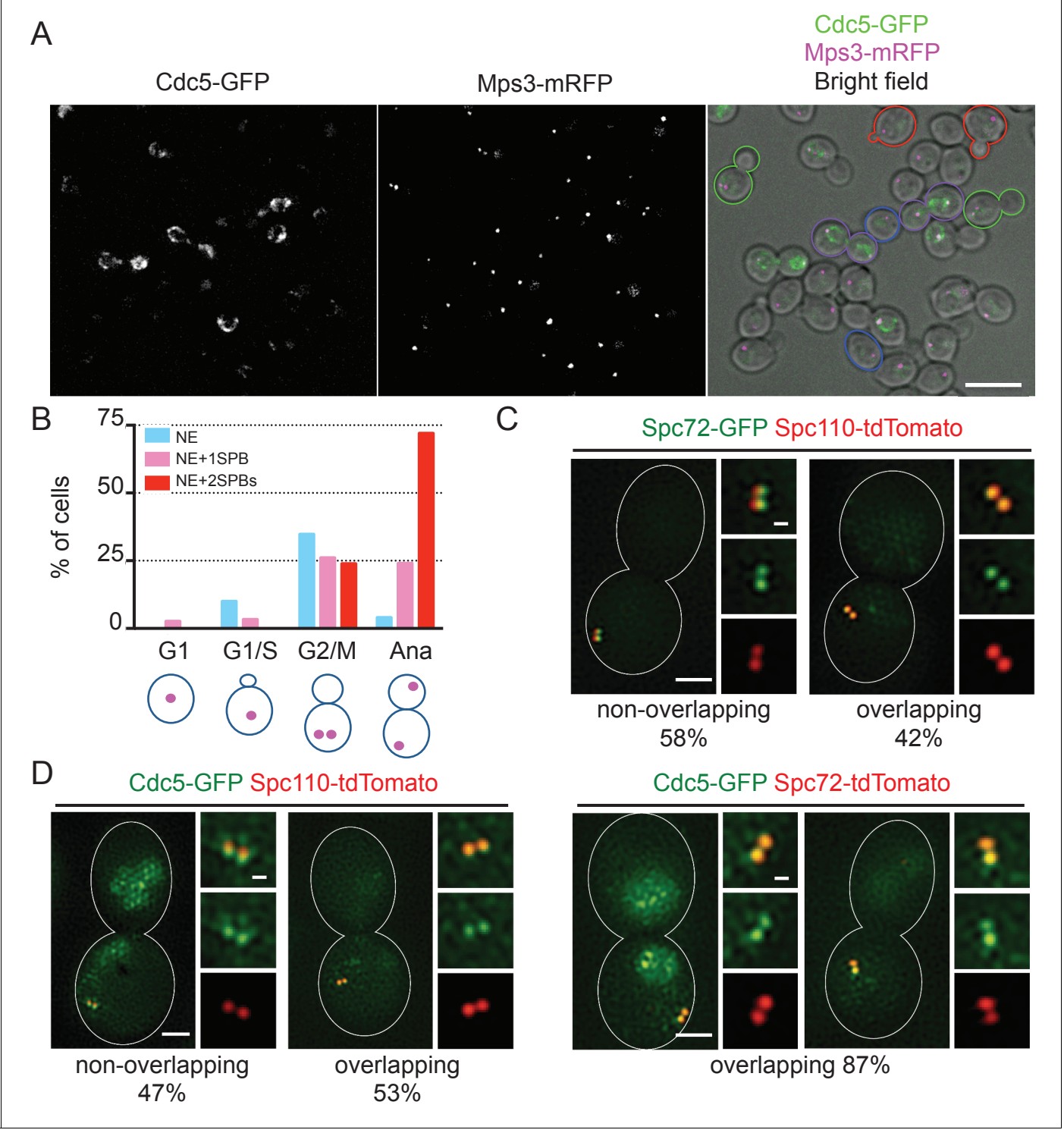

**Figure 7.** Localization of polo-like Cdc5 kinase in the nucleus, the nuclear envelope, and at the SPB is cell cycle dependent. (**A**) *CDC5-GFP MPS3-mRFP* cells (HPH1562) were grown in YPDS at 30°C. Blue, red, green, and purple cell contours mark G1, S/G2, G2/M, and anaphase cells, respectively. Shown is a projected image after deconvolution. Scale bar, 5 μm. (**B**) Quantification of Cdc5-GFP localization in (**A**). The position of SPB (magenta) was as outlined in the cartoon shown at the bottom of the subfigure. n > 30 cells for each cell cycle stage. Ana, anaphase. (**C**) SIM images of Spc72-GFP and Spc110-tdTomato in nocodazole-arrested cells. *SPC72-GFP SPC110-tdTomato* cells (HPH1581) grown in SC medium at 30°C were arrested in metaphase with nocodazole. Scale bars, 1 μm and 0.2 μm in the large and small images, respectively. n = 78. (**D**) SIM images of Cdc5-GFP together with either Spc110-tdTomato or Spc72-tdTomato in nocodazole-arrested cells. *CDC5-GFP SPC110-tdTomato* and *CDC5-GFP SPC72-tdTomato* cells

*Figure 7 continued on next page*

*Figure 7 continued*

grown in SC medium at 30°C were arrested in metaphase with nocodazole. Scale bars, 1 µm and 0.2 µm in the large and small images, respectively. Strains used were HPH1583 (n = 75) and HPH1575 (n = 55). Diffused nuclear signal of Cdc5-GFP were also observed in all cells (*Figure 7—figure supplement 2*).

DOI: https://doi.org/10.7554/eLife.24340.030

The following figure supplements are available for figure 7:

**Figure supplement 1.** Localization of Cdc5-GFP.
DOI: https://doi.org/10.7554/eLife.24340.031
**Figure supplement 2.** Localization of Cdc5-GFP.
DOI: https://doi.org/10.7554/eLife.24340.032

here that the ascomycetous yeast *O. polymorpha* does not follow the same strategy. In *O. polymorpha*, the nucleus generally locates centrally within the mother cell body and the spindle is not aligned properly along mother-bud axis until anaphase onset. Consequently, spindle elongation in early anaphase occurs entirely in the mother often with an inappropriate angle against the polarity axis. Despite this potential complication, one nucleus penetrates successfully into the bud during anaphase, which may largely rely on an immediate correction of the orientation of the spindle and on SPOC activity. Those SPB movements are in contrast to *S. cerevisiae* in which spindle is aligned during metaphase and therefore SPB translocation into the bud coincides with spindle elongation. Currently molecular mechanism(s) that regulate spindle orientation is unknown. However, although the timing of spindle orientation relative to cell cycle progression appears to be different from that of other yeasts, two redundant molecular mechanisms of spindle orientation, one requiring dynein and the other Kar9, may be conserved in *O. polymorpha*, because putative orthologs of *KAR9* and dynein were identified in *O. polymorpha* genome sequences (*Li et al., 1993*; *Miller and Rose, 1998*; *Maekawa and Kaneko, 2014*; *Nordberg et al., 2014*).

In *S. cerevisiae*, Spc72 is stably incorporated into SPBs once it is recruited and organise cMTs throughout the cell cycle. In *O. polymorpha*, the strong SPB association of OpSpc72 in anaphase becomes weakened as cells enter into the following G1 phase, whereas they re-accumulate later in the cell cycle. The timing of OpSpc72 recruitment to SPBs during early mitosis appears to primarily dictate the organization of cMTs and hence nuclear position (*Figure 9*). As the polo-like kinase Cdc5 protein plays an important role in this regulation, a question arises regarding the substrates of Cdc5 kinase in this process (*Archambault and Glover, 2009*). Because ScSpc72 binds to and is phosphorylated by Cdc5 in *S. cerevisiae*, OpSpc72 represents an obvious candidate (*Maekawa et al., 2007*; *Snead et al., 2007*). Our electrophoresis analyses indeed detected Cdc5-dependent phosphorylation of Spc72. However, OpCdc5 failed to phosphorylate recombinant Spc72 in vitro. Further analyses are required to verify whether Cdc5 directly phosphorylates Spc72 and the potential effects of such modifications on the regulation of cMT. Additionally, it may also be important to identify other kinase(s) that are responsible for the Cdc5-independent phosphorylation of Spc72 observed in our analyses. Cdc5 may have another important substrates other than Spc72. Notably, a polo box binding site present in ScSpc72 is missing in OpSpc72. Cdc5 might therefore phosphorylate other SPB proteins such as Nud1, which has one site matching the consensus sequences of polo box binding site (S-Sp/Tp-P), and thereby indirectly influence the affinity of Spc72 towards the SPB.

It is unclear what brought highly expressed Spc72-GFP to SPBs at early cell cycle stages when Cdc5 activity was low. Weak Spc72-GFP signal in the absence of Cdc5 may be because of low affinity or unstable association to SPBs. Increased Spc72 protein levels may simply result in the increased number of Spc72 protein at SPBs at any given time. Alternatively, overexpression may overcome a negative regulation that normally maintains the low level of Spc72 at SPBs during early stages of the cell cycle (*Figure 4*). Such Cdc5-independent regulation is consistent with the observation that Spc72 was gradually lost from SPBs at early stages of the cell cycle. Examining properties of Spc72 protein when overexpressed such as post-translational modifications, protein stability, and molecular dynamics at SPBs would clarify this point.

In *S. cerevisiae*, SPB duplication initiates in late G1 phase by forming a satellite at the distal end of an extended half-bridge of the pre-existing 'old' SPB, which is then inserted into the nuclear envelope. In contrast to the old SPB which maintains cMTs from the previous mitosis, this 'new' SPB acquires the MT nucleation activity in the inner plaque prior to spindle assembly while it is still

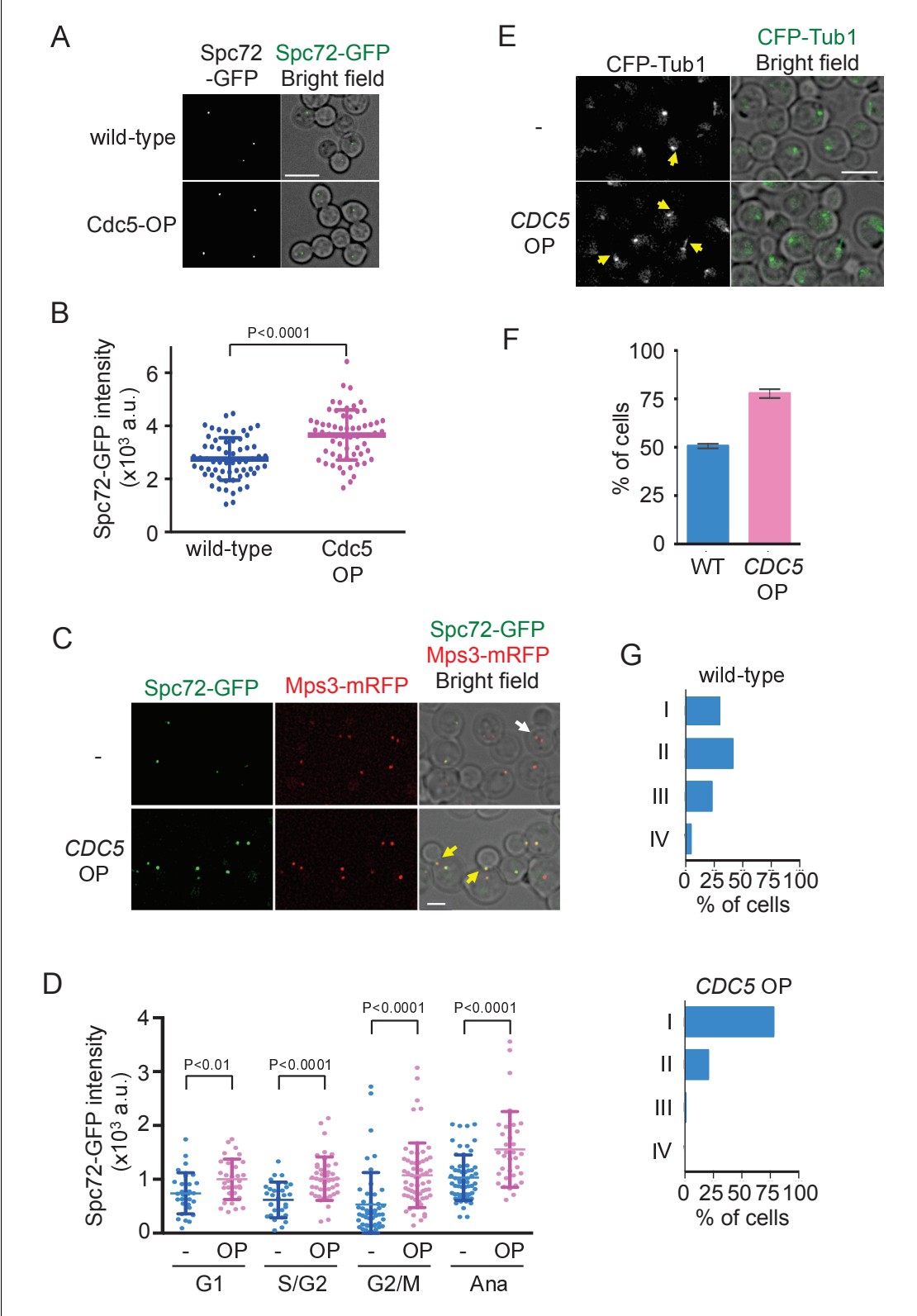

**Figure 8.** Overexpression of Cdc5 kinase promotes early association of Spc72 to SPBs. (**A**) Overexpression of *CDC5* enhances SPB binding of Spc72. *SPC72-GFP MPS3-mRFP* (HPH1394) and *SPC72-GFP MPS3-mRFP P*$_{TEF1}$*-CDC5Δ53* cells (HPH1542) were arrested in metaphase with nocodazole. Deconvolved and projected images are shown. Scale bar, 5 µm. (**B**) Spc72-GFP signal at SPBs in (**A**) was quantified. Signal intensities were background-subtracted. Error bars indicate SD. n > 50 cells. (**C**) Spc72-GFP was recruited to SPB at earlier stages of the cell cycle when *CDC5* gene is

*Figure 8 continued on next page*

*Figure 8 continued*

overexpressed. *SPC72-GFP MPS3-mRFP* (HPH1394, -) and *SPC72-GFP MPS3-mRFP P_{TEF1}-CDC5Δ53* cells (HPH1542, *CDC*5 OP) were grown in YPDS medium. Enrichment of Spc72 to SPB and alignment of the spindle along the mother-bud axis were evident in preanaphase cells (yellow arrows) overexpressing *CDC5*, compared with wild-type cells (white arrow). Shown are deconvolved and projected images. Mps3-mRFP marks SPB. Scale bar, 2 μm. (D) Quantification of Spc72-GFP intensity at SPBs in (C). –: wild-type; OP: *CDC5* overexpression. Signal intensities were background-subtracted. Statistical significances of the difference between wild-type (-) and *CDC5* overexpressing cells (*CDC5* OP) were determined by the student t-test. Error bars indicate SD. (E) Overexpression of Spc72 stimulated cMT acquisition. MTs were visualized with *CFP-TUB1* in wild type and *CDC5* overexpressing cells (HPH1680 and HPH1673, respectively) were grown in YPDS medium at 30°C. Images were captured only for CFP and brightfield. Scale bar, 5 μm. (F) Quantification of (E). Presence/absence of cMTs was scored in cells with short spindle. Shown is the average of three independent experiments. Error bars indicate SD. n > 100. (G) SPB is positioned close to the bud neck in G2/M cells overexpressing *CDC5*. The position of the SPB closer to the bud was as outlined in the cartoon shown in *Figure 5B*. Strains used were HPH1394 (n = 125) and HPH1542 (n = 101).
DOI: https://doi.org/10.7554/eLife.24340.033

The following figure supplements are available for figure 8:

**Figure supplement 1.** Spc72-GFP protein level was not affected by *CDC5* overexpression.
DOI: https://doi.org/10.7554/eLife.24340.034
**Figure supplement 2.** Preanaphase spindle was aligned along the mother-bud axis in cells overexpressing *CDC5*.
DOI: https://doi.org/10.7554/eLife.24340.035

connected to the old SPB (side-by-side SPBs). On the other hand, a recent report suggested that the acquisition of ScSpc72 to the outer plaque, and hence of cMTs, occurs only after SPB separation and spindle assembly (*Juanes et al., 2013*). In *O. polymorpha,* the cMTs acquisition is also regulated at the level of Spc72 recruitment to SPBs. OpSpc72 dissociate from SPBs at the end of mitosis and recruited to both old and new SPBs shortly before anaphase of the following cell cycle. This suggests that the cell cycle regulation of Spc72 recruitment may be applied to both old and new SPBs in *O. polymorpha*. Even though the Spc72 recruitment is cell cycle regulated in both species, its timing seems to be different: while it occurs in early G2 phase in *S. cerevisiae*, it does in metaphase in *O. polymorpha*. Moreover, their regulatory mechanisms are likely different because their acquisition is Cdc5-dependent in *O. polymorpha*, but not in *S. cerevisiae* (*Juanes et al., 2013*). Inhibition of Cdc5 kinase in *S. cerevisiae* causes the misaligned spindle phenotype, which indicates a role of Cdc5 in cMT functions (*Snead et al., 2007*). However, it is unlikely that Cdc5 acts at the level of Spc72 recruitment since both SPBs of the misaligned spindle caused by Cdc5 inhibition carried cMTs (*Snead et al., 2007*). Alternatively, it is possible that the Cdc5-dependent regulation of ScSpc72

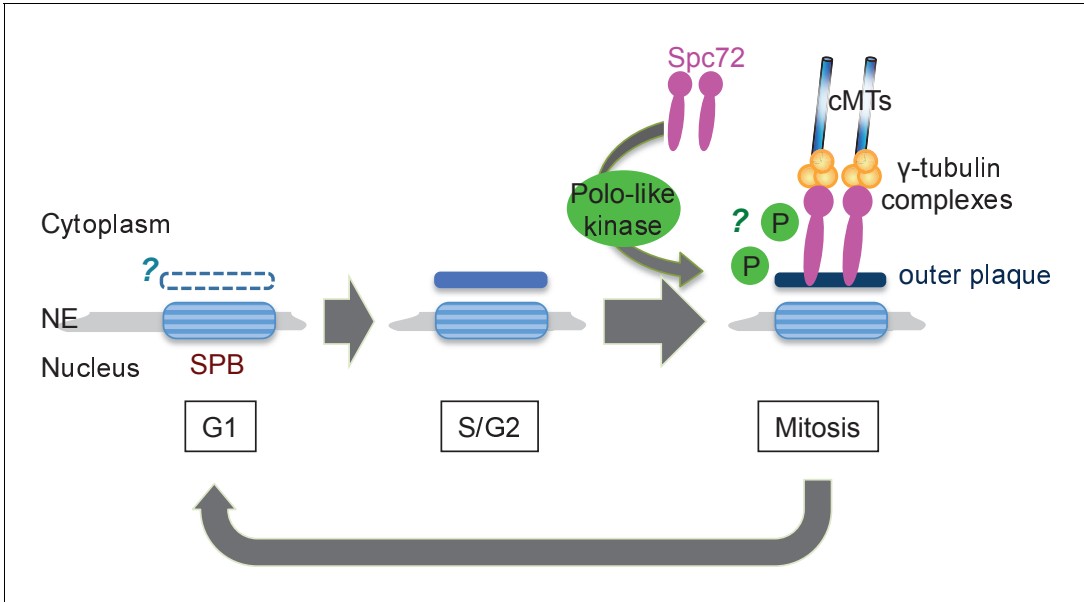

**Figure 9.** Model of the SPB cycle in *O. polymorpha*. See Discussion for details.
DOI: https://doi.org/10.7554/eLife.24340.036

function on cMT nucleation/anchoring has been overlooked. Growth suppression of *cnm67Δ* cells by *CDC5* overexpression might point towards this possibility (*Park et al., 2004*). It might also be because of the redundancy with the SPB-anchored Stu2 function (*Usui et al., 2003*). The Stu2 binding domain of Spc72 (aa176–230 in ScSpc72, *Figure 4—figure supplement 5*) is conserved in ascomycetous yeasts including *O. polymorpha* but it differs somewhat in OpSpc72. In particular, the detached cMTs observed in approximately 14% of *O. polymorpha* pre-anaphase cells are reminiscent of the *spc72^{ΔStu2}* phenotype in *S. cerevisiae* (*Usui et al., 2003*). Thus, the potential binding of Stu2 to Spc72 in *O. polymorpha* should be investigated.

In *O. polymorpha*, structure of the cytoplasmic side of SPB, along with cMT nucleation competence, is modified as the cell cycle progresses (*Figure 9*). Notably, our BLAST search failed to identify orthologs of several SPB core components identified in *S. cerevisiae* in genomes of outside of Saccharomycetaceae: the central plaque components Spc42 and Spc29, the membrane anchors Nbp1 and Bbp1, and the half-bridge component Kar1. The failure of identification may be due to the highly divergent nature of amino acid sequences of these coiled-coil proteins, or alternatively the function of these SPB components is not required in *O. polymorpha*. The absence of these proteins could be one of the reasons underlying the poor appearance of the central plaque and the half-bridge in our electron microscopy anlayses. Our observation that the outer plaque in the cytoplasm was evident in anaphase SPBs but not in G1 SPBs may suggest the outer plaque from the previous mitosis is removed in the following G1 phase. However, given that Nud1, a putative outer plaque component, is present at the SPB in G1, it is more likely that the outer plaque is only partially disassembled. Furthermore, the appearance of the outer plaque in EM analysis proceeded that of Spc72 in fluorescent microscopy (*Figure 9*). Electron microscopy analysis using another fixation method that better preserves the SPB structure might clarify this point. Equally important is the analysis of the half-bridge which organizes cMTs in G1 in *S. cerevisiae*. If the half-bridge plays the same role in *O. polymorpha* SPB as that in *S. cerevisiae*, the loss of Spc72 from outer plaque alone should not cause the loss of cMTs. In addition, in *S. cerevisiae*, SPBs are segregated in a defined mode where the old SPB normally migrates into the daughter cell while the new SPB remains in the mother cell (*Pereira et al., 2001*). The SPB history in the outer plaque was proposed to primarily determine the destination of SPBs and to bias spindle asymmetry via Nud1 (*Hotz et al., 2012*; *Juanes et al., 2013*). Whether the mode of SPB inheritance is conserved in *O. polymorpha* and other yeast species is also worthy of further study.

It is unlikely that the mode of nuclear positioning in *O. polymorpha* is ancestral, given that *Y. lipolytica* and *P. pastoris,* who are diverted from the common ancestor earlier than *O. polymorph*, share the same nuclear organization with *S. cerevisiae*. However, several yeast species relatively close to *O. polymorpha* or *C. albicans* exhibited a similar nuclear position in pre-anaphase cells (*Figure 1—figure supplement 1*). Thus, the Spc72 recruitment mechanism described in *O. polymorpha* may be widely utilized in several Clades in Saccharomycotina, including *Ogataea*, *Ambrosiozyma*, and *Nakazawaea*.

## Materials and methods

### Yeast strains and plasmids

Yeast strains and plasmids used in this study are listed in *Table 1*. Unless otherwise indicated, all *O. polymorpha* strains were derived from NCYC 495 and were generated by PCR-based methods (*Lu et al., 2000*; *Janke et al., 2004*; *Saraya et al., 2012*). GFP, mRFP, and 5flag tagged alleles were generated in *ku80Δ* or *ku70Δ* cells and then crossed with auxotrophic wild-type strains to obtain *KU80^+* or *KU70^+* cells carrying the tagged allele (*Maekawa and Kaneko, 2014*). *O. polymorpha* cells were transformed by electroporation (*Faber et al., 1994*). The 500 bp sequences up- and downstream of the *OpTEF1* open reading frame (ORF) were used as the *OpTEF1* promoter and terminator, respectively and those of the *OpCDC28* ORF were used as the *OpCDC28* promoter and terminator (*Kiel et al., 2007*). For overexpression of *CDC5*, we expressed an N-terminal-truncated version of *CDC5* (*CDC5Δ53*) that is equivalent to the *ScCDC5ΔN70* allele which is resistant to APC-dependent ubiquitination (*Shirayama et al., 1998*). The *CDC5Δ53* ORF was amplified by PCR and inserted into pHM949. The resulting plasmid pHM950 was digested and a zeocin resistance marker was inserted (pHM956). To obtain the *dyn1Δ* strain, tetrad dissection plates were incubated at room temperature for >5 days until colonies were formed because the *dyn1Δ* cells grew extremely slowly. Glycerol stocks

**Table 1.** Yeast strains and plasmids

| Name | Genotype/species/construction | Source or reference |
|---|---|---|
| O. polymorpha strains | | |
| BY4329 | leu1-1 | NBRP |
| BY21401 | CBS4732 Type strain | NBRP |
| HPH31 | HHT1::pHM713 ura3-1 | this study |
| HPH41 | ura3-1 pHM719 | this study |
| HPH164 | HHT1::pHM726 TUB1:: pHM737 leu1-1 | this study |
| HPH194 | TUB1::pHM737 leu1-1 | this study |
| HPH206 | Δdyn1::natNT2 leu1-1 | this study |
| HPH207 | Δdyn1::natNT2 leu1-1 | this study |
| HPH221 | wild type | this study |
| HPH222 | Δbub2::hphNT1 | this study |
| HPH223 | Δkar9::natNT2 | this study |
| HPH224 | Δkar9::natNT2 Δbub2::hphNT1 | this study |
| HPH225 | leu1-1 | this study |
| HPH399 | SPC72-GFP-hphNT1 | this study |
| HPH449 | SPC72-mRFP-hphNT1 TUB1:: pHM737 leu1-1 | this study |
| HPH466 | wild type | this study |
| HPH475 | hphNT1-PSPC98-GFP-SPC98 | this study |
| HPH972 | SPC72-GFP-hphNT1 | this study |
| HPH1150 | Δcdc28::natNT2::pHM878 ura3-1 | this study |
| HPH1210 | Δcdc28::natNT2::pHM878 TUB1:: pHM737 HHT1::pHM713 ura3-1 | this study |
| HPH1380 | SPC72-GFP-hphNT1 CDC5-3mAID-natNT2 ura3-1::pHM922 | this study |
| HPH1393 | SPC72-GFP-hphNT1 MPS3-mRFP-kanMX6 ura3-1::pHM859 | this study |
| HPH1394 | SPC72-GFP-hphNT1 MPS3-mRFP-kanMX6 | this study |
| HPH1396 | hphNT1-PNUD1-GFP-NUD1 | this study |
| HPH1400 | SFI1-GFP-hphNT1 | this study |
| HPH1405 | SPC110-GFP-hphNT1 | this study |
| HPH1430 | SPC72-5flag-hphNT1 CDC5-3mAID-natNT2 ura3-1::pHM922 | this study |
| HPH1542 | SPC72-GFP-hphNT1 MPS3-mRFP-kanMX6 TEF1::pHM950::pHM956 | this study |
| HPH1562 | CDC5-GFP-hphNT1 MPS3-mRFP-kanMX6 | this study |
| HPH1564 | CDC5-GFP-hphNT1 MPS3-mRFP-kanMX6 HHT1::pHM726 | this study |
| HPH1575 | CDC5-GFP-hphNT1 SPC72-tdTomato-hphMX | this study |
| HPH1581 | SPC72-tdTomato-hphMX SPC110-GFP-hphNT1 | this study |
| HPH1583 | CDC5-GFP-hphNT1 SPC110-tdTomato-natNT2 | this study |
| HPH1678 | MPS3-GFP-hphNT1 HHT1::pHM713 | this study |
| other yeast strains | | |
| BY21467 | S. cerevisiae YPH499 | NBRP |
| BY21165 | Kluyveromyces lactis NH27 | NBRP |
| BY21167 | Yarrowia lipolytica T22 | NBRP |
| BY23876 | Candida glabrata YAT3377 | NBRP |
| BY5243 | Ogataea parapolymorpha DL-1 | NBRP |
| JCM9829 | Candida peltata | JCM |
| JCM 10237 | Ogataea methanolica | JCM |
| JCM15019 | Ambrosiozyma kashinagacola | JCM |

*Table 1 continued on next page*

*Table 1 continued*

| Name | Genotype/species/construction | Source or reference |
| --- | --- | --- |
| plasmids | | |
| pHM713 | pREMI-Z carrying HHT1-GFP and HHF1(histoneH4) | this study |
| pHM719 | pKS144 carrying TUB4-GFP | this study |
| pHM726 | pREMI-Z carrying HHT1-mCherry and HHF1(histoneH4) | this study |
| pHM737 | pRS305 carrying PTUB1-GFP-TUB1 | this study |
| pHM859 | pBSII carring HpURA3 and PTEF1-HpSPC72-GFP | this study |
| pHM878 | pBSII carring HpURA3 and cdc28-as | this study |
| pHM922 | pBSII carring HpURA3 and PCDC28-OsTIR | this study |
| pHM950 | pFA6a-natNT2 carrying PTEF1-HpCDC5Δ53 | this study |
| pHM956 | pFA6a-natNT2 carrying PTEF1-HpCDC5Δ53 and zeo | this study |

DOI: https://doi.org/10.7554/eLife.24340.037

were prepared from the initial master plate of tetrad analysis. YPDS liquid medium was inoculated with either the initial colonies from tetrad dissection or glycerol stocks, and the resulting cells were subjected to analyses.

## Yeast growth conditions and general methods

Yeast strains were grown either in YPD medium containing 200 mg/l adenine, leucine, and uracil (YPDS) or in synthetic/defined (SD) medium supplemented with appropriate amino acids and nucleotides (*Sherman, 1991*). Cells were grown at 30℃ unless otherwise indicated. To depolymerize MTs, cells were incubated in either YPDS medium or SD medium containing 1.5 μg/ml nocodazole at 30℃ for 1.5 hr.

## Microscopy

Yeast cells carrying *GFP-TUB1*, *HHT1-mCherry*, *SPC72-GFP*, *GFP-NUD1*, *SFI1-GFP*, *SPC98-GFP*, *TUB4-GFP*, *SPC110-GFP*, *CDC5-GFP*, or *MPS3-mRFP* were immediately analysed by fluorescence microscopy without washing or fixation in *Figures 1A*, *2*, *4*, *5*, *6B–E*, *7* and *8A*, *Figure 4—figure supplement 1*, *Figure 5—figure supplement 1*. For the visualisation of DNA with 4'6,-diamidino-2-phenylindole (DAPI), cells were fixed with 70% ethanol, washed with phosphate buffered saline (PBS), and incubated in PBS containing 1 μg/ml DAPI.

Z-series images of 0.4 μm steps were captured with DeltaVision (Applied Precision, Issaquah, WA, USA) equipped with GFP and TRITC filters (Chroma Technology Corp., Bellows Falls, VT, USA), a 100 × NA 1.4 UPlanSApo oil immersion objective (IX71; Olympus, Tokyo, Japan), and a camera (CoolSNAP HQ; Roper Scientific, Trenton, NJ, USA) and were quantified/processed with SoftWoRx 3.5.0 (Applied Precision, Issaquah, WA, USA) or Prism4.3.0 software (*Chen et al., 1992*; *Chen et al., 1996*). Deconvolved and projected images are shown. The fluorescence intensity of Spc72-GFP was measured on a plane that has an SPB in focus. Time-lapse experiments of *Figures 1A* and *2D* and that of *Figure 4C* were carried out in YPDS and SD complete medium respectively on a glass-bottom dish (MatTek, Ashland, MA, USA) coated with concanavalin A (037–08771, Wako, Japan) at room temperature. Z series at 0.4 μm steps were acquired every 3 min for *Figures 1A* and *4C*, or every 30 s for *Figure 2D*.

For *Figure 1B*, cells were fixed with 70% ethanol, washed with PBS, and incubated in PBS containing 1 mg/ml DAPI to visualize the DNA (*Maekawa and Kaneko, 2014*). ImageJ 1.47 (NIH, Bethesda, MD, USA) and Photoshop (Adobe Systems, San Jose, CA, USA) were used to mount the images and to produce merged colour images. No manipulations other than contrast and brightness adjustments were used.

To exclude cells that were non-proliferating from the GFP intensity measurements in *Figure 4B*, cells were first incubated in YPDS containing Alexa 594 conjugated concanavalin A (Thermo Fisher Scientific, Waltham, MA, USA) until all cells were labelled and then washed once with YPDS and

incubated in YPDS for 1 hr prior to image capture. Cells that had lost the label or had a bud with no label were subject to the analyses.

## Electron microscopy

Cells were mounted on a glass-bottom dish (MatTek) coated with concanavalin A and covered with fixative [2% glutaraldehyde in 0.1 M sodium phosphate buffer (pH 7.2)]. After 1 min, cells were further fixed with fresh fixative for 2 hr at 4°C. After washing with buffer, low melting agarose was applied onto the cells to prevent loss of cells during subsequent procedures. Zymolyase solution (0.4 mg/ml zymolyase 100T, Seikagaku Co., Tokyo, Japan) was applied on top of the agarose for 60 min at 37°C, postfixed with 2% $OsO_4$ for 2 hr at room temperature, stained with 1% uranyl acetate for 1 hr, dehydrated with acetone in an ascending series from 50% to 100%, and embedded in epoxy resin. Serial sections of 80 nm thicknesses were obtained, poststained with uranyl acetate and lead citrate, and analysed using a Zeiss EM900 Transmission Electron Microscope at Central Unit Electron Microscopy in the German Cancer Research Center (DKFZ) (ZEISS, Oberkochen, Germany) or a Hitachi H-7500 Transmission Electron Microscope at Research Centre for Ultra-High Voltage Electron Microscopy at Osaka University (Hitachi, Tokyo, Japan).

## Cell cycle analysis and growth conditions

For synchronization, *CDC5-3mAID* cells were incubated in YPDS containing indole-3-acetic acid (IAA) (45533, Sigma-Aldrich, St. Louis, MO, USA)) for 2.5 hr at 30°C until >80% of cells had a large sized bud to deplete Cdc5 (*Nishimura and Kanemaki, 2014*). Cells were then washed with pre-warmed growth medium to remove IAA and re-suspended in YPAD medium at 30°C.

## Yeast cell extract and immunoblotting

Whole cell extracts were prepared for SDS-PAGE and immunoblotting (*Knop et al., 1999*; *Janke et al., 2004*; *Meitinger et al., 2016*). Samples representing 1–2 OD600 of liquid culture were resuspended in 950 µl 0.29 M NaOH and incubated on ice for 10 min. Then, 150 µl 55% (w/v) tri-chloroacetic acid was added and the solutions were mixed and incubated for 10 min on ice. After centrifugation the supernatant was removed. The protein pellet was resuspended in high urea buffer (8 M urea, 5% SDS, 200 mM $NaPO_3$ pH 6.8, 0.1 mM EDTA, 100 mM dithiothreitol, and bromophenol blue) and heated at 65°C for 10 min. A sample comprising one-fifth of the total sample amount was loaded for SDS-PAGE (*Figure 6A*) and western blotting was performed using a standard protocol. For immunoprecipitation, total cell extracts were prepared from logarithmically growing cells in immunoprecipitation buffer (100 mM Tris, pH 8.0, 10 mM EDTA, 150 mM NaCl, 5% glycerol, 0.2 mM $NaVO_3$100 mM β-glycerophosphate, 50 mM NaF, 1 mM PMSF, 1 mM DTT, 1% NP-40, and Complete EDTA-free protease inhibitor cocktail [Roche]). 10 mg of total cell extract was incubated with M2-bound magnetic beads (M8823, Sigma) for 2 hr at 4°C. The beads were washed three times with immunoprecipitation buffer. The bound proteins were subjected to λ phosphatase treatment and then eluted in 30 µl of SDS-PAGE sample buffer by incubated at 37°C for 30 min. 3 µl of eluates were loaded on a Mini-PROTEAN TGX Precast Gels (4561021, BIO-RAD Laboratories, Hercules, CA, USA) and western blotting was performed using a standard protocol. Monoclonal antibodies JL-8 (632381, TaKaRa Bio Clontech, Shiga, Japan) and M2 (F1804, Sigma) were used to detect GFP- and flag-tagged proteins respectively. Plot profile function of ImageJ was used to plot intensity value across a line in *Figure 6—figure supplement 2*.

## Structured illumination microscopy (SIM)

Cells were arrested for 2.5 hr with 1.5 µg/ml nocadazole and fixed for 15 min in 4% paraformaldehyde/2% sucrose in phosphate-buffered saline (PBS) solution followed by extensive washing in PBS. The cells were immobilized on a concanavalin A (Sigma-Aldrich, MO, USA)- coated 35 mm glass bottom dish (MatTek, P35G-1.5–14C) and maintained in PBS for the duration of the imaging process in PBS. The samples were imaged in the 2D-SIM mode on a Nikon N-SIM system (Tokyo, Japan) equipped with a TIRF Apochromat 100x/1.49 NA oil immersion objective and a single photon detection EM-CCD camera (Andor iXon3 DU-897E; Belfast, UK). The 488 nm and 561 nm laser lines were used for excitation of yeGFP and tdTomato, respectively, combined with emission band pass filter 520/45 and 610/60. Images were taken sequentially within a small z-stack and in consideration of

imaging SPBs close to the coverslip to minimise spherical aberrations. Subsequently the reconstruction and channel alignment was performed using the NIS imaging and image analysis software (Nikon). For the xyz chromatic shift correction we used in a reference sample tetraspeck beads in a reference sample. All images show a single stack of the z-slices.

## Acknowledgements

We thank Dr. K Richter and the Central Unit Electron Microscopy at the German Cancer Research Center (DKFZ) for their kind support for the electron microscopy (EM) analyses. The Nikon Imaging Facility Heidelberg is acknowledged for the SIM analyses. We thank Research Center for Ultra-High Voltage Electron Microscopy at Osaka University for supporting our EM analyses. We thank Japan Collection of Microorganisms (JCM) and Dr T Endo for yeast strains, information, and advice on phylogenetic relationships among yeast species. We thank Dr. T Lin for assistance on bioinformatics. Dr. H Takuma, Dr. Yamaguchi, and Integrated Imaging Research Support (IISR) Japan are acknowledged for technical advice and the primary EM analysis, respectively. HM is grateful to Dr. Takegawa for generously allowing access to facilities in the lab. This work was supported by the Endowed Chair Program of the Institute for Fermentation, Osaka (IFO), Japan (YK) and JSPS KAKENHI Grant Number JP24570214 (HM). GP acknowledges funding from the German Research Council (DFG, PE1883) and DFG collaborative programs SFB873 and SFB1036. ES acknowledges funding from the German Research Council (DFG, Schi 295/4-3).

## Additional information

### Funding

| Funder | Grant reference number | Author |
|---|---|---|
| Japan Society for the Promotion of Science | JP24570214 | Hiromi Maekawa |
| Deutsche Forschungsgemeinschaft | Schi 295/4-3 | Elmar Schiebel |
| Deutsche Forschungsgemeinschaft | PE1883 | Gislene Pereira |
| Deutsche Forschungsgemeinschaft | SFB873 | Gislene Pereira |
| Deutsche Forschungsgemeinschaft | SFB1036 | Gislene Pereira |
| Institute for Fermentation, Osaka | the Endowed Chair Program | Yoshinobu Kaneko |

The funders had no role in study design, data collection and interpretation, or the decision to submit the work for publication.

### Author contributions

Hiromi Maekawa, Conceptualization, Resources, Data curation, Formal analysis, Funding acquisition, Investigation, Methodology, Writing—original draft, Project administration, Writing—review and editing; Annett Neuner, Diana Rüthnick, Investigation, Writing—review and editing; Elmar Schiebel, Gislene Pereira, Supervision, Writing—review and editing; Yoshinobu Kaneko, Resources, Funding acquisition

### Author ORCIDs

Hiromi Maekawa (iD) http://orcid.org/0000-0002-0175-1610
Elmar Schiebel (iD) http://orcid.org/0000-0002-3683-247X
Gislene Pereira (iD) http://orcid.org/0000-0002-6519-4737
Yoshinobu Kaneko (iD) https://orcid.org/0000-0002-7379-9373

**Decision letter and Author response**
Decision letter https://doi.org/10.7554/eLife.24340.043
Author response https://doi.org/10.7554/eLife.24340.044

## Additional files

### Supplementary files
• Transparent reporting form
DOI: https://doi.org/10.7554/eLife.24340.038

### Major datasets
The following previously published datasets were used:

| Author(s) | Year | Dataset title | Dataset URL | Database, license, and accessibility information |
|---|---|---|---|---|
| Maekawa H, Kaneko Y | 2014 | Opolymorpha_4329 | https://www.ncbi.nlm.nih.gov/bioproject/PRJDB3035 | Submitted to the DNA Data Bank of Japan and publicly available under BioProject ID PRJDB3035 |
| Riley R, Haridas S, Wolfe KH, Lopes MR, Hittinger CT, Goker M, Salamov AA, Wisecaver JH, Long TM, Calvey CH, Aerts AL, Barry KW, Choi C, Clum A, Coughlan AY, Deshpande S, Douglass AP, Hanson SJ, Klenk HP, LaButti KM, Lapidus A, Lindquist EA, Lipzen AM, Meier-Kolthoff JP, Ohm RA, Otillar RP, Pangilinan JL, Peng Y, Rokas A, Rosa CA, Scheuner C, Sibirny AA, Slot JC, Stielow JB, Sun H, Kurtzman CP, Blackwell M, Grigoriev IV, Jeffries TW | 2016 | Ogataea polymorpha NCYC 495 leu1.1 v2.0 | http://genome.jgi.doe.gov/Hanpo2/Hanpo2.home.html | The U.S. Department of Energy is committed to making its electronic and information technologies accessible to individuals with disabilities in accordance with Section 508 of the Rehabilitation Act (29 U.S.C. 794d), as amended in 1998. |

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
