## [Decision Letter]

Thank you for submitting your article "Polo-like kinase Cdc5 regulates Spc72 recruitment to spindle pole body in the methylotrophic yeast *Ogataea polymorpha*" for consideration by *eLife*. Your article has been reviewed by three peer reviewers, one of whom is a Guest Reviewing Editor for *eLife*, and the evaluation has been overseen by Anna Akhmanova as the Senior Editor. The reviewers have opted to remain anonymous.

The reviewers have discussed the reviews with one another and the Reviewing Editor has drafted this decision to help you prepare a revised submission.

Summary:

The orientation of the mitotic spindle axis determines the direction of chromosome segregation in a spatiotemporal manner. In many eukaryotic cells undergoing close mitosis such as yeast, intranuclear spindle positioning is controlled by cytoplasmic microtubules (cMTs). A similar role is played by astral microtubules in open mitosis.

In their manuscript, Maekawa et al. undertake the first characterisation of spindle positioning in the budding yeast *O. polymorpha*. The work reveals outstanding differences in the sequence of events previously established in *S. cerevisiae*, a well-studied paradigm. These idiosyncrasies are invaluable for dissecting cell cycle controls that are more apparent in one particular system while nevertheless conserved, as they converge on the spindle pole body or SPB, the yeast functional equivalent of the centrosome. Maekawa et al. present evidence for unbiased nuclear position (and the spindle within) in *O. polymorpha* at early stages of the cell cycle. However, the elongating spindle aligns along the mother-bud axis and inserts into the bud neck during anaphase. By contrast, the spindle in *S. cerevisiae* adopts a polarised orientation much earlier via prominent cMTs. Accordingly, Maekawa et al. observed a very low number of cMTs from G1 to anaphase onset in *O. polymorpha*. The authors linked this distinction to the late addition of Spc72 (a γ-tubulin complex receptor) to the SPB cytoplasmic face, under the control of the polo-like kinase Cdc5. Supporting this view, depletion or overexpression of Cdc5 abolished or anticipated Spc72 recruitment, respectively. Furthermore, cells overexpressing Cdc5 preempted nuclear migration close to the bud neck, pointing to Spc72 recruitment as a possible trigger for spindle positioning.

The reviewers have concurred that these correlations constitute important and interesting findings. However, the key claim that cell cycle-dependent recruitment of Spc72 controls cMT organisation and nuclear positioning in *O. polymorpha* should be supported by direct evidence and not solely left to be inferred from those correlations.

Therefore, as detailed below, the reviewers recommend that:

– direct observation of cMTs should be carried out to verify their active involvement in nuclear/spindle positioning along the cell cycle.

– conclusive evidence supporting the nature of the link between Cdc5 and Spc72 should be provided.

Essential revisions:

1) Since this is a first characterisation, an overview of the spindle pathway should be documented by providing:

– A set of representative images for an SPB marker and the nucleus at the different stages along the entire cell cycle.

– Time lapse data of cells expressing an SPB marker to document the kinetics of SPB separation in order to define discrete stages by length along the spindle pathway, as previously done in *S. cerevisiae*. Stages such as "large budded preanaphase cells" (e.g. Results section first paragraph) should be correctly defined in quantitative terms.

2) The authors should report the impact of deleting *SPC72*.

3) Experiments were performed exclusively in vegetative cells. Are Spc72 recruitment and cMT organisation affected by nutrient conditions that induce mating? Are cMTs expected to have a role during mating?

There might be key differences regarding the (half)bridge with respect to *S. cerevisiae* (Figure 3). This should be stated clearly for its implications in cMT organisation along the cell cycle. Could the authors provide more detailed EM analysis? In particular, does SPB duplication proceed along the lines shown for *S. cerevisiae*? Is the (half)-bridge visible at any point before or after duplication? Is there a satellite in late G1? Could you provide corresponding images for *S. cerevisiae* SPBs obtained under identical conditions to appreciate differences in outer plaque appearance?

4) Nuclear position in *O. polymorpha* is described as random in the Abstract and main text. Yet, the nucleus tends to position centrally (Results section first paragraph and Figure 1). Time lapse must be presented to cover the events leading to this central position using cells expressing GFP-Tub1 for the interval between mitotic exit (when nuclei are positioned at opposite ends by the spindle) and the emergence of the bud in the following cell cycle to draw conclusions regarding any relationship between the axis of cell polarity and SPB/nuclear position. Please clarify and amend the text accordingly.

5) Subsection “*O. polymorpha* cells contain only fewer cMTs”, concerning cMT detachment. It is unclear how cMTs that dissociate from the SPB arise. Are cMTs indeed "unstable"? Time-lapse analysis should be implemented to document in more detail dynamic instability parameters as well as the timeline from cMT emergence to detachment.

6) Figure 2 and Figure 2—figure supplement 1. The cMTs in the time-lapse series are very hard to visualize. Arrows pointing at cMTs should be added and the basis for quantification of cMT persistence and cMT re-establishment should be fully described so that the significance of these data is made clear.

7) Figure 4. It is unclear when Spc72 level changes along the preanaphase interval. Is label lost after a short spindle forms? Furthermore, there are issues of consistency throughout the manuscript regarding images and quantification of Spc72-GFP signals. In Figure 4 cells do not show Spc72-GFP signal. Yet, the quantification in Figure 4 points to a higher signal in G1 than in S/G2. Again, in Figure 4 at time 0 signal is absent despite the quantification shown before. Later on, in Figure 8, quantification of Spc72-GFP at different stages of the cell cycle indicates lower signals than in Figure 4. It is important to explain this variability and show the quantification of several replicates of the experiments.

In Figure 4, the legend describes "signal intensities were background-subtracted". This may not be a proper method. Instead, Mps3-mRFP signals should be used for internal controls. This is also the case for Figure 6.

With regards to Figure 4, unfortunately the interval of live imaging, i.e. 3 min, is too long to visualise the timing of the increase in Spc72-GFP signals during the cell cycle.

Time lapse analysis of cells expressing Spc72-GFP should be extended to document how the label is lost at the short spindle stage and regained close to anaphase as hinted by still image data, using suitable temporal resolution (at least 1 min intervals).

8) On overproduction of Spc72 (Figure 5). The authors claim that by overproducing Spc72, SPBs become localised to the vicinity of the bud neck in preanaphase cells. It is critical to observe cMTs. Please show MT structures, possibly by time-lapse live imaging.

9) In Figure 6, cells were synchronised using an inducible degron to deplete Cdc5. However, in support of the authors' claims, time course analysis following release from a nocodazole block should be carried out to monitor Spc72 levels, localisation and phosphorylation in the next cell cycle in otherwise wild type cells.

Regarding phosphorylation of Spc72 (Figure 6—figure supplement 2). The difference between the lanes is too subtle to conclude on the specific contribution of Cdc5. The experiment should be performed using Phos-tag gels or changing the acrylamide/bis ratio to better display Spc72 phosphoisoforms. An alternative mode of synchronisation could be implemented, since nocodazole arrest is likely to further enhance phosphorylation thus masking the specific contribution of Cdc5. Finally, the authors should provide data to address the physical interaction between Cdc5 and Spc72 and in vitro kinase assays to demonstrate whether Spc72 is a Cdc5 substrate.

10) On Cdc5 localisation (Figure 7). Using SIM, the authors claim that Cdc5 is localised exclusively to the cytoplasmic side of the SPB during metaphase. Are the authors claiming that Cdc5-polo kinase plays no roles inside the nucleus during mitosis in *O. polymorpha*? Please clarify this issue.

Along the same lines, most SPB markers in *S. cerevisiae* contribute cytoplasmic label by a free pool, the same holds true for Cdc5. The images in this manuscript virtually lack any whole cell background – this suggests over-processing. Diffuse nuclear label might have been lost as well here. The authors should provide sample raw images to assess whether loss of whole-cell and nuclear signals might be due to processing.

11) Regarding Cdc5 overproduction (Figure 8). The impact of manipulating Cdc5 on cMT presence should be assessed directly in cells expressing GFP-Tub1 to score cMTs along the cell cycle. In wild type or Cdc5 overexpressors, when do cMT plus ends access the bud?

Figure 8 presents an increase in Spc72-GFP signal rather than advanced recruitment of Spc72 along the cell cycle. This rise might be due to increased Spc72 protein level and/or phosphorylation. Controls should be added to report on Spc72 levels and modification upon Cdc5 overexpression. Moreover, Cdc5 localisation upon overexpression should be analysed to determine whether Cdc5 association to the SPB is also affected.

Finally, it remains unclear whether Cdc5 kinase activity or only Cdc5 association to the SPB is the key determinant for Spc72 recruitment. While SPB tethering experiments might help resolve this point, the authors should at least address whether the ability of Cdc5 to recruit Spc72 to the SPB may involve their physical interaction. To this end, co-immunoprecipitation assays or protein pull downs with recombinant baits should be provided.

12) Could the authors propose how Spc72 acquisition brings about spindle alignment? At least they should provide time lapse data of cells expressing GFP-Tub1 at sufficient temporal resolution to visualise cMT acquisition and the dynamics of spindle alignment and insertion across the bud neck to judge the involvement of cMTs in response to Spc72 loading.

13) Figure legends should be edited for clarity and plots should be described in full for what is being scored in every case. Arrows or arrowheads should be added in every image to point at relevant structures that are difficult to discern without this aid as requested above.

[Editors' note: further revisions were requested prior to acceptance, as described below.]

Thank you for resubmitting your work entitled "Polo-like kinase Cdc5 regulates Spc72 recruitment to spindle pole body in the methylotrophic yeast *Ogataea polymorpha*" for further consideration at *eLife*. Your revised article has been favorably evaluated by Anna Akhmanova (Senior editor), and a Guest Reviewing editor.

The manuscript has been improved but there are some remaining issues that need to be addressed by editing the text of the manuscript before acceptance, as outlined below.

The authors have addressed the bulk of the original review but left some specific points unanswered, mainly on the grounds of technical limitations in the use of *O. polymorpha* as a model. Given this problem, those points should be qualified correctly in the final report.

Taken together, the manuscript portrays substantial novelty if focused on the behaviour of Spc72, Cdc5 and cMTs. Thus, it would be important to carry out a number of revisions in the text for the narrative in the Results and Discussion to be brought in line with the level of proof. Following is a list of action points and we would like to encourage the authors to revise accordingly so that data are not mis-represented here.

1) Concerning the path for spindle orientation in *O. polymorpha*

Although nuclear position is not biased toward the bud neck, SPB movement depicted in the time lapse series for SPB localisation is inconsistent with the statement that position is entirely random (e.g. Discussion first paragraph), as it is steadily maintained at the mother cell centre. The temporal resolution used in this study may not reveal the underlying basis for this position but it does not convey proof of random location either. Thus, the text should be amended here to reflect the actual data.

Without detracting from the key conclusion on the role of Spc72 recruitment, we note that the representative time lapse series presented as supplemental data to Figure 1 showed the following: a) In Figure 1—figure supplement 4, after assembly 1 µm long spindles remain centrally positioned and loosely oriented toward the bud neck b) According to Figure 1—figure supplement 5, spindle alignment is apparent at onset of spindle elongation followed later by SPB translocation. Instead, in *S. cerevisiae*, SPB translocation (of spindles held aligned during metaphase) coincides with spindle elongation. The analysis and narration of the current Figure 1 mis-represents this dynamic behaviour. We would recommend moving the time lapse series to be part of the main Figure 1 and, if space is limiting, move supporting quantitation of still images to the supplemental section of Figure 1. Furthermore, the narration should be edited to reflect these data faithfully.

2) On Dyn1 and Kar9 pathways and SPOC function in *O. polymorpha* (final paragraph in subsection “Nuclear positioning in *O. polymorpha* differs from that in *S. cerevisiae* and other budding yeast species” and second paragraph of the Discussion section) the data are not sufficiently developed to make these claims and should be substantially edited for the following reasons:

– In the new sample image of dyn1∆ cells (Figure 1—figure supplement 6) it is very difficult to tell the nuclear stain from the cell background and hard to see how the quantitation was possible based on such data. Furthermore the stain is inconsistent with the live image of nuclear label by tagging. These data do not support conclusions regarding nuclear size as a differential factor between *S. cerevisiae* and *O. polymorpha*. Finally, the defect in the dyn1∆ mutant may transcend cMT function per se in line with the severe growth defect. Note that dyn1∆ mutant cells appeared to re-bud inappropriately and accumulate supernumerary nuclei, arguing that the SPOC may not operate as robustly (if at all) compared to *S. cerevisiae*, resulting in low viability of dyn1∆ cells. Furthermore, the interaction between kar9∆ and bub2∆ appears much stronger than what is apparent in *S. cerevisiae* (in which kar9∆ bub2∆ is not lethal at 30^°^C)

Under the circumstances, the authors are merely inferring cellular defects that they have not proven satisfactorily, for inclusion in this report. The same holds true for their claim that the Kar9 and Dynein pathways are functionally conserved between the two yeasts. cMT images failed to show plus ends reaching the distal bud cell cortex to talk about Kar9-dependent positioning here.

In conclusion, this section should be deleted from a revised manuscript to maintain the focus on the findings linking cMTs and Spc72. Also note that the authors could not answer here a number of essential queries from the original review on technical grounds.

– Discussion paragraph two. Please revise to simply mention synthetic lethality between kar9 and dyn1 mutants, with dyn1 alone showing a severe growth defect at best. Whether this is due to the demands of aligning the spindle during anaphase (as opposed to *S. cerevisiae* in which spindle alignment is achieved prior to anaphase) remains highly speculative. The severe growth defect due to dyn1∆ in *O. polymorpha* is consistent with weak SPOC-dependent delay. As stated above, the authors report excess of fragmented nuclei and re-budding in dyn1∆ cells, supporting this idea.

3) The definition of metaphase spindle is not kept consistent throughout. At one point, metaphase is defined by SPB distance < 2 µm but earlier in the text, it was defined as < 1 µm. Please amend accordingly.

4) The progressive and gradual loss of Spc72 from SPBs at early stages of the cell cycle is inconsistent with Cdc5 acting as the sole switch in Spc72 recruitment (with reversal triggered by mitotic exit). The authors should make this point clearer when discussing Spc72 detection in G1 cells in Figure 4. Please revise to bring the narration in line with the data.

5) In the phosphorylation analysis, the change of tags on Spc72 should be explained in full and justified.

6) The rational for testing Spc72 upon starvation could be made clearer – mating is triggered by starvation in *O. polymorpha*.

7) Discussion paragraph three, on Nud1 being a possible Cdc5 target – please clarify if you meant that Nud1 contains a consensus site for binding the Cdc5 polo box (rather than a polo box consensus sequence?).

8) The authors should discuss why Spc72 overexpression is sufficient to force recruitment to SPBs in their model (data in Figure 5).

9) The Introduction is incorrect regarding Spc72 and cMTs in *S. cerevisiae*. Simply put, cMTs at the new SPB are formed after a 1 µm long spindle has formed (certainly not G1/S!, Shaw et al., 1997). Please revise to state correctly that the new SPB acquires Spc72 and cMTs after onset of spindle assembly (also see Juanes et al., 2013).

---

## [Author Response]

Essential revisions:1) Since this is a first characterisation, an overview of the spindle pathway should be documented by providing:– A set of representative images for an SPB marker and the nucleus at the different stages along the entire cell cycle.– Time lapse data of cells expressing an SPB marker to document the kinetics of SPB separation in order to define discrete stages by length along the spindle pathway, as previously done in S. cerevisiae. Stages such as "large budded preanaphase cells" (e.g. Results section first paragraph) should be correctly defined in quantitative terms.

According to the reviewer’s comment, we now analyzed SPB behavior in respect to bud formation and/or chromosome segregation. Representative images of cells with SPB marker is shown in Figure 1. Time lapse analysis is presented in Figure 1—figure supplement 4 and Figure 1—figure supplement 5.

Comparison of Mps3-GFP intensity in cells with different bud size suggested SPB duplicates in small budded cells as shown in *S. cerevisiae*. Duplicated and separated SPBs upon the short spindle formation maintain close proximity until anaphase onset, as shown in Figure 1—figure supplement 4. Anaphase was initiated in cells with large bud and the event occurred within the mother cell body (Figure 1—figure supplement 5). These results are described in paragraph two of subsection “Nuclear positioning in *O. polymorpha* differs from that in S. cerevisiae and other budding yeast species” of the revised manuscript, and “pre-anaphase cells” are now defined at the end of the paragraph.

2) The authors should report the impact of deleting SPC72.

We analyzed the essentiality of *SPC72* for growth by tetrad analysis using a *SPC72/spc72Δ::natNT2* heterozygous diploid strain. We found that *SPC72* is essential for growth. The result is shown in Figure 4—figure supplement 2 and described in subsection “Spc72 associates with SPB in a cell cycle-dependent manner”.

3) Experiments were performed exclusively in vegetative cells. Are Spc72 recruitment and cMT organisation affected by nutrient conditions that induce mating? Are cMTs expected to have a role during mating?

This is a very interesting point. We now analysed Spc72 localisation in starved conditions. We found that while cMTs were not observed, Spc72-mRFP or Spc72-GFP accumulated at SPBs under starvation conditions (representative images of *SPC72-mRFP GFP-TUB1* cells are shown in Figure 5—figure supplement 2). This may suggest that Spc72 recruited to SPBs does not immediately organize cMTs in such conditions. Although we assume that cMTs are involved in karyogamy, we were unable to analyse cMTs in zygotes because mating frequency was low and zygotes failed to proceed karyogamy once they were placed on a glass-bottom dish. We are now discussing these data in the final paragraph of “Spc72 associates with SPB in a cell cycle-dependent manner”.

There might be key differences regarding the (half)bridge with respect to S. cerevisiae (Figure 3). This should be stated clearly for its implications in cMT organisation along the cell cycle. Could the authors provide more detailed EM analysis? In particular, does SPB duplication proceed along the lines shown for S. cerevisiae? Is the (half)-bridge visible at any point before or after duplication? Is there a satellite in late G1? Could you provide corresponding images for S. cerevisiae SPBs obtained under identical conditions to appreciate differences in outer plaque appearance?

We agree that clarifying the structure and cell cycle-dependency of half-bridge and bridge formation is an important issue. In *S. cerevisiae*, EM studies of satellite formation at the half-bridge have been facilitated by the use of synchronised cultures. Pheromone induced G1 arrest (similar to α-factor synchronisation in *S. cerevisiae*) is not yet established in *O. polymorpha*. Thus, we tried alternative strategies. We constructed *cdc28-as* strain in order to observe SPB in late G1. But *cdc28-as* cells did not arrest at the same cell cycle stage as in *S. cerevisiae*. We think that this is because the inhibitor cannot completely inhibit Cdc28-as kinase. Nevertheless, we observed a delay in cell cycle progression (accumulation of cells with a single SPB signal by fluorescence microscopy) upon Cdc28-as inhibition. We could show by EM that some of these cells had side-by-side SPBs, which suggested that SPB duplication proceed along the lines shown for *S. cerevisiae*. We could detect an electron dense cloud between the two SPBs on the cytoplasmic side of the nuclear envelope (Figure 3—figure supplement 2BE). Therefore, we think that a bridge-like structure is also formed in *O. polymorpha*. However, it was not clear whether the half-bridge/bridge like structure was present at other cell cycle stages. These data is presented in Figure 3—figure supplement 2, and the description of the result is in paragraph two of subsection “Organization of the SPB structure on the cytoplasmic side is cell cycle dependent”.

In subsection “Organization of the SPB structure on the cytoplasmic side is cell cycle dependent”, we also added following sentence “while half-bridge-like structure, which plays an important role in cMT organization in G1 of *S. ccerevisiae*, was not clearly observed”.

In the Discussion section, possible difference of the half-bridge/bridge is described: “Equally important is the analysis of the half-bridge which, in *S. cerevisiae,* organizes cMTs in G1. If the half-bridge plays the same role in *O. polymorpha* SPB as that in *S. cerevisiae*, the loss of Spc72 from outer plaque alone should not cause the loss of cMTs”.

Also, earlier timing of outer plaque appearance than Spc72-GFP is stated in the Discussion: “Furthermore, the appearance of the outer plaque in EM analysis proceeded that of Spc72 in fluorescent microscopy (Figure 9).”

4) Nuclear position in O. polymorpha is described as random in the Abstract and main text. Yet, the nucleus tends to position centrally (Results section first paragraph and Figure 1). Time lapse must be presented to cover the events leading to this central position using cells expressing GFP-Tub1 for the interval between mitotic exit (when nuclei are positioned at opposite ends by the spindle) and the emergence of the bud in the following cell cycle to draw conclusions regarding any relationship between the axis of cell polarity and SPB/nuclear position. Please clarify and amend the text accordingly.

We think that because nucleus is a large structure relative to the cell size, it is more likely to be located near the cell centre than at the periphery in the absence of external forces. Thus, we do not think that there is a specific regulation for positioning the nucleus centrally. To clarify this point, time lapse movies of SPB movement during anaphase to G1 of the next cell cycle are presented as Figure 1—figure supplement 5. In these movie series, we could see the rapid and continuous movement of SPBs in the entire cell body. Thus, we concluded that SPB position is random in G1. The result is described in the text: “After spindle breakdown, the SPB moved vigorously in no relationship to the polarity axis (Figure 1—figure supplement 5).”

5) Subsection “O. polymorpha cells contain only fewer cMTs”, concerning cMT detachment. It is unclear how cMTs that dissociate from the SPB arise. Are cMTs indeed "unstable"? Time-lapse analysis should be implemented to document in more detail dynamic instability parameters as well as the timeline from cMT emergence to detachment.

We have now performed Time lapse microscopy using cells expressing *GFP-TUB1* and *HHT1-mCherry*. We see that cMTs dettached from SPBs and remained in the cytoplasm for only a short period of time before disappearing. This data thus indicate that detached cMT are short-lived. We could not determine dynamic instability parameters due to fast disappearance of detached cMTs. Therefore, to avoid confusion, we replaced “unstable” with “short-lived”.

Representative images of time lapse series showing cMT detachment are shown in Figure 2—figure supplement 1, and the result is described in the text: “Time lapse analysis revealed that detached cMTs remained in the cytoplasm only for a short period of time before depolymerized (Figure 2—figure supplement 1).”

6) Figure 2 and Figure 2—figure supplement 1. The cMTs in the time-lapse series are very hard to visualize. Arrows pointing at cMTs should be added and the basis for quantification of cMT persistence and cMT re-establishment should be fully described so that the significance of these data is made clear.

Thanks for pointing this out. For better visualization of cMTs, we have replaced one of the time lapse sequences with another example in which cMTs are easily visible in Figure 2—figure supplement 3. Yellow arrows are added to point cMTs.

7) Figure 4. It is unclear when Spc72 level changes along the preanaphase interval. Is label lost after a short spindle forms? Furthermore, there are issues of consistency throughout the manuscript regarding images and quantification of Spc72-GFP signals. In Figure 4 cells do not show Spc72-GFP signal. Yet, the quantification in Figure 4 points to a higher signal in G1 than in S/G2. Again, in Figure 4 at time 0 signal is absent despite the quantification shown before. Later on, in Figure 8, quantification of Spc72-GFP at different stages of the cell cycle indicates lower signals than in Figure 4. It is important to explain this variability and show the quantification of several replicates of the experiments.

We are sorry for this confusion. Decrease of Spc72-GFP SPB signal after mitotic exit is slow and the timing varied from cell to cell. We think this may be the reason why some of G1 cells carry strong Spc72-GFP signal. Time-lapse series from anaphase to the next cell cycle are now shown in Figure 4—figure supplement 4, and the results are described in the text: “As cells exit from mitosis and entre into the next cell cycle, Spc72-GFP signal was gradually decreased at SPBs with the timing that varied from cell to cell. However, in all cases, Spc72-GFP levels reached a minimum well before short spindle was formed (Figure 4 and Figure 4—figure supplement 4).”

In Figure 4, the legend describes "signal intensities were background-subtracted". This may not be a proper method. Instead, Mps3-mRFP signals should be used for internal controls. This is also the case for Figure 6.

In all quantification, signal intensities were background-subtracted. We do not think that Mps3-mRFP is a suitable internal control in *O. polymorpha* because Mps3-mRFP signal bleached so fast that its intensity decline even during z-series acquisition. There is no other SPB components that gives stronger RFP signal than Mps3. Therefore we think that the background-subtraction is currently the best method available.

With regards to Figure 4, unfortunately the interval of live imaging, i.e. 3 min, is too long to visualise the timing of the increase in Spc72-GFP signals during the cell cycle.Time lapse analysis of cells expressing Spc72-GFP should be extended to document how the label is lost at the short spindle stage and regained close to anaphase as hinted by still image data, using suitable temporal resolution (at least 1 min intervals).

We now show new time lapse series of Spc72-GFP with 30 sec or 1 min intervals are shown in Figure 4 and Figure 4—figure supplement 3 and Figure 4—figure supplement 4.

The text is changed to “In all cells that progressed into anaphase, an Spc72-GFP signal became detectable < 4 min prior to the initiation of anaphase (average 3.68 ± 1.74 min, n=14) (Figure 4 yellow arrowhead). Within 5 min after appearance of the Spc72-GFP signal, spindle orientation was corrected when it had not done already (Figure 4—figure supplement 3, average 3.50 ± 1.61 min, n=12); therefore, one half part of an anaphase nucleus was successfully inserted into the bud.”

8) On overproduction of Spc72 (Figure 5). The authors claim that by overproducing Spc72, SPBs become localised to the vicinity of the bud neck in preanaphase cells. It is critical to observe cMTs. Please show MT structures, possibly by time-lapse live imaging.

According to the reviewer’s comment, we visualized cMTs by introducing *CFP-TUB1* in SPC72-overexpressing cells (Figure 5). cMTs are now observed more frequently and the following sentence was added; “cMTs are more often observed (Figure 5)”.

9) In Figure 6, cells were synchronised using an inducible degron to deplete Cdc5. However, in support of the authors' claims, time course analysis following release from a nocodazole block should be carried out to monitor Spc72 levels, localisation and phosphorylation in the next cell cycle in otherwise wild type cells.

In our hands, the only synchronization protocol that arrest cells and then allow them to resume the cell cycle was the Cdc5-depletion/re-expression. Other methods did not work. For examples, although *O. polymorpha* cells can be arrested by nocodazole, they cannot be easily released. Even at the lowest concentration that causes cell cycle arrest, cells required more than two hours to resume cell cycle and did so in an unsynchronized fashion. We generated degron strains for *CDC4, MCM4, MCM10* as well as *cdc28-as*, but none of them worked well enough for synchronization.

In order to address the fluctuation of Spc72 protein level in cell cycle, we have compared Spc72 protein levels in cycling cells, nocodazole arrested cells (metaphase), *cdc28-as* cells (delayed G1/S phase), *CDC5-3mAID* cells (late anaphase). We found that the amount of Spc72 protein was at the comparable levels in all cells. The result is presented in Figure 6—figure supplement 1 and described in the text; “Spc72 protein abundance did not fluctuate as cells entered into anaphase and proceeded into the following cell cycle (Figure 6 and Figure 6—figure supplement 1).”

As for the localization of Spc72 from anaphase to the next cell cycle, we performed time lapse analysis in asynchronized *SPC72-GFP* cells. Although we could not obtain time lapse series that covers the entire cell cycle due to bleaching of GFP signal, we could show that Spc72-GFP disappeared as cells exit mitosis and before the bud emergence of the following cell cycle. The results are presented in Figure 4 (4.5 min and 17 min) and Figure 4—figure supplement 4. Appearance of Spc72-GFP prior to anaphase onset is presented in Figure 4 min) and Figure 4—figure supplement 3 (5.5 min). The results are described in subsection “Spc72 associates with SPB in a cell cycle-dependent manner”

We have compared phosphorylation status of Spc72 in asynchronous and nocodazole-arrested cells and found that it migrated slower in nocodazole-arrested cells. This result suggests Spc72 is subjected to cell cycle dependent phosphorylation. The result is now presented in Figure 6—figure supplement 1 and described in the text: “Furthermore, the Spc72 band migrated slower in nocodazole-arrested cells than that in asynchronous cells (Figure 6—figure supplement 1).”

Regarding phosphorylation of Spc72 (Figure 6—figure supplement 2). The difference between the lanes is too subtle to conclude on the specific contribution of Cdc5. The experiment should be performed using Phos-tag gels or changing the acrylamide/bis ratio to better display Spc72 phosphoisoforms. An alternative mode of synchronisation could be implemented, since nocodazole arrest is likely to further enhance phosphorylation thus masking the specific contribution of Cdc5. Finally, the authors should provide data to address the physical interaction between Cdc5 and Spc72 and in vitro kinase assays to demonstrate whether Spc72 is a Cdc5 substrate.

We performed western blotting analysis to better show Cdc5-dependent phosphorylation of Spc72 using total protein prepared under a denatured condition (Figure 6—figure supplement 3). In our hand, Phos-tag gels did not help resolving Cdc5-dependent phosphorylation.

As for physical interaction between Cdc5 and Spc72, we did in vitro pull-down assay using recombinant Spc72 as a bait, but the result was inconclusive because of non-specific binding between Cdc5 and sepharose beads (data not shown). We also performed in vitro kinase assay with negative results. However, we cannot completely exclude that Spc72 is not a direct substrate of Cdc5. Previously, we have shown that *S. cerevisiae* Cdc5 can phosphorylate recombinant Spc72, but incorporation of 32P was much stronger for the degradation products of the recombinant Spc72 (Maekawa et al., 2007). This may suggest that a specific conformation or SPB binding is necessary for the reaction. Whether Spc72 is directly phosphorylated by Cdc5, is therefore still an open question. This point is further discussed in the text: “Cdc5 may have another important substrates other than Spc72. Notably, a polo box binding site present in ScSpc72 is missing in OpSpc72. Cdc5 might therefore phosphorylate other SPB proteins such as Nud1, which has one site matching the polo box consensus sequences (S-Sp/Tp-P), and thereby indirectly influence the affinity of Spc72 towards the SPB.”

10) On Cdc5 localisation (Figure 7). Using SIM, the authors claim that Cdc5 is localised exclusively to the cytoplasmic side of the SPB during metaphase. Are the authors claiming that Cdc5-polo kinase plays no roles inside the nucleus during mitosis in O. polymorpha? Please clarify this issue.

We thank the reviewer to point it out. We did observe nuclear signal of Cdc5-GFP and assume that it has nuclear functions during the cell cycle. To clarify this, we added the following sentence: “Nuclear and NE localisation appeared at early stages of the cell cycle and persisted until the end of mitosis”. Also we modified the text; “Thus, Cdc5 likely becomes first localised to the nucleus and the NE in G2, and then in mitosis to the cytoplasmic side of SPBs.”

The legend of Figure 7 now clearly states the nuclear localization of Cdc5-GFP.

Along the same lines, most SPB markers in S. cerevisiae contribute cytoplasmic label by a free pool, the same holds true for Cdc5. The images in this manuscript virtually lack any whole cell background – this suggests over-processing. Diffuse nuclear label might have been lost as well here. The authors should provide sample raw images to assess whether loss of whole-cell and nuclear signals might be due to processing.

We thank the reviewer to point it out. We prepared the high intensity version of SIM images of Cdc5-GFP, in which diffuse nuclear signal is easily visible. It is present in Figure 7—figure supplement 2.

11) Regarding Cdc5 overproduction (Figure 8). The impact of manipulating Cdc5 on cMT presence should be assessed directly in cells expressing GFP-Tub1 to score cMTs along the cell cycle. In wild type or Cdc5 overexpressors, when do cMT plus ends access the bud?

In order to address this point, cMTs were visualized by introducing *CFP-TUB1* in *CDC5*-overexpressing cells (Figure 8). CFP-Tub1 signal was not strong enough to unambiguously determine the timing of cMT contact with the bud cortex. However, cMTs were more frequently observed. This is now discussed in the text: “and cMTs were more prevalent (Figure 8).”

Figure 8 presents an increase in Spc72-GFP signal rather than advanced recruitment of Spc72 along the cell cycle. This rise might be due to increased Spc72 protein level and/or phosphorylation. Controls should be added to report on Spc72 levels and modification upon Cdc5 overexpression. Moreover, Cdc5 localisation upon overexpression should be analysed to determine whether Cdc5 association to the SPB is also affected.

We have added data in Figure 8—figure supplement 1 showing that Spc72-GFP protein levels were not affected by Cdc5 depletion or *CDC5*-overexpression. The result is described in subsection “CDC5 overexpression accelerates the Spc72 recruitment to SPB”; “While Cdc5 expression showed no effect on the protein level of Spc72-GFP (Figure 8—figure supplement 1),”

Finally, it remains unclear whether Cdc5 kinase activity or only Cdc5 association to the SPB is the key determinant for Spc72 recruitment. While SPB tethering experiments might help resolve this point, the authors should at least address whether the ability of Cdc5 to recruit Spc72 to the SPB may involve their physical interaction. To this end, co-immunoprecipitation assays or protein pull downs with recombinant baits should be provided.

In order to address this point, we have overexpressed kinase-dead version of *CDC5* to see whether physical presence of the protein stimulate the recruitment of Spc72 to SPB. However, this experiment did not bring a conclusion because overexpressed kinase-dead Cdc5 did not localize to SPB at any stage of the cell cycle.

We attempted another experiments to address this issue, as previously discussed in comment 9. However, what determines Spc72 recruitment to SPB is still an open question.

12) Could the authors propose how Spc72 acquisition brings about spindle alignment? At least they should provide time lapse data of cells expressing GFP-Tub1 at sufficient temporal resolution to visualise cMT acquisition and the dynamics of spindle alignment and insertion across the bud neck to judge the involvement of cMTs in response to Spc72 loading.

In order to address this point, we performed timelapse analysis of *GFP-TUB1* cells with short intervals. We could show the correlation between cMT acquisition and the spindle orientation. The result is presented in Figure 2—figure supplement 3 and described in the text; “Acquired cMTs efficiently corrected the spindle orientation in pre-anaphase cells, suggesting that the spindle orientation is regulated largely at the level of cMT acquisition (Figure 2—figure supplement 3).”

Time lapse data showing the relative timing between Spc72 loading and the spindle orientation is presented in Figure 4 and Figure 4—figure supplement 3 and the result is discussed in the text; “Within 5 min after appearance of the Spc72-GFP signal, spindle orientation was corrected when it had not done already (Figure 4—figure supplement 3, average 3.50 ± 1.61 min, n=12); therefore, one half part of an anaphase nucleus was successfully inserted into the bud.”

Time lapse with double labelling of Spc72 and MTs was technically not feasible in *O. polymorpha*. Spc72-RFP signal beached out too quickly for time lapse and mCherry-Tub1 was not efficiently incorporated into microtubule filaments.

13) Figure legends should be edited for clarity and plots should be described in full for what is being scored in every case. Arrows or arrowheads should be added in every image to point at relevant structures that are difficult to discern without this aid as requested above.

Figure legends are modified to include all required information. Relevant structures and fluorescent signals are now marked with arrows/arrowheads.

[Editors' note: further revisions were requested prior to acceptance, as described below.]

Taken together, the manuscript portrays substantial novelty if focused on the behaviour of Spc72, Cdc5 and cMTs. Thus, it would be important to carry out a number of revisions in the text for the narrative in the Results and Discussion to be brought in line with the level of proof. Following is a list of action points and we would like to encourage the authors to revise accordingly so that data are not mis-represented here.1) Concerning the path for spindle orientation in O. polymorphaAlthough nuclear position is not biased toward the bud neck, SPB movement depicted in the time lapse series for SPB localisation is inconsistent with the statement that position is entirely random (e.g. Discussion first paragraph), as it is steadily maintained at the mother cell centre. The temporal resolution used in this study may not reveal the underlying basis for this position but it does not convey proof of random location either. Thus, the text should be amended here to reflect the actual data.

The text is now modified to “the nucleus generally locates centrally”.

Without detracting from the key conclusion on the role of Spc72 recruitment, we note that the representative time lapse series presented as supplemental data to Figure 1 showed the following: a) In Figure 1—figure supplement 4, after assembly 1 µm long spindles remain centrally positioned and loosely oriented toward the bud neck b) According to Figure 1—figure supplement 5, spindle alignment is apparent at onset of spindle elongation followed later by SPB translocation. Instead, in S. cerevisiae, SPB translocation (of spindles held aligned during metaphase) coincides with spindle elongation. The analysis and narration of the current Figure 1 mis-represents this dynamic behaviour. We would recommend moving the time lapse series to be part of the main Figure 1 and, if space is limiting, move supporting quantitation of still images to the supplemental section of Figure 1. Furthermore, the narration should be edited to reflect these data faithfully.

One of time lapse series in the former Figure 1—figure supplement 5 is now moved to Figure 1 as Figure 1. To accommodate the time lapse in Figure 1, quantification data (the former Figure 1) have been moved to a supplement (Figure 1—figure supplement 3).

a) Passage on the still images of SPB marker and the time lapse in Figure 1DE and Figure 1—figure supplement 6 have been modified in the text; “Moreover, SPB in G1 cells as well as small budded cells was not in the defined position within the mother cell body (Figure 1). Subsequent time lapse analysis revealed that after spindle assembly, ~ 1 µm long spindles remained at their central positions and loosely oriented toward the bud neck until shortly before anaphase onset.”

b) Time lapse images in Figure 1 and Figure 1—figure supplement 6 represent cells with mis-aligned spindle. However, there are cells with loosely positioned/aligned spindle as shown in Figure 5 (wild type category I and II), and it is not possible to narrow down the precise timing of spindle alignment. Therefore, we left the time window for spindle alignment wider than “anaphase onset” as suggested by the reviewer, and added the following passage to the Results; “Spindle alignment was corrected around the time of (or shortly after) spindle elongation, followed by SPB insertion into the bud.”

We have also added the following texts; “after spindle assembly, ~ 1 µm long spindles remain at their central positions and loosely oriented toward the bud neck”, and “Those SPB movements are in contrast to *S. cerevisiae* in which spindle is aligned during metaphase and therefore SPB translocation into the bud coincides with spindle elongation.”

2) On Dyn1 and Kar9 pathways and SPOC function in O. polymorpha (final paragraph in subsection “Nuclear positioning in O. polymorpha differs from that in S. cerevisiae and other budding yeast species” and second paragraph of the Discussion section) the data are not sufficiently developed to make these claims and should be substantially edited for the following reasons:– In the new sample image of dyn1∆ cells (Figure 1—figure supplement 6) it is very difficult to tell the nuclear stain from the cell background and hard to see how the quantitation was possible based on such data. Furthermore the stain is inconsistent with the live image of nuclear label by tagging. These data do not support conclusions regarding nuclear size as a differential factor between S. cerevisiae and O. polymorpha. Finally, the defect in the dyn1∆ mutant may transcend cMT function per se in line with the severe growth defect. Note that dyn1∆ mutant cells appeared to re-bud inappropriately and accumulate supernumerary nuclei, arguing that the SPOC may not operate as robustly (if at all) compared to S. cerevisiae, resulting in low viability of dyn1∆ cells. Furthermore, the interaction between kar9∆ and bub2∆ appears much stronger than what is apparent in S. cerevisiae (in which kar9∆ bub2∆ is not lethal at 30^°^C)Under the circumstances, the authors are merely inferring cellular defects that they have not proven satisfactorily, for inclusion in this report. The same holds true for their claim that the Kar9 and Dynein pathways are functionally conserved between the two yeasts. cMT images failed to show plus ends reaching the distal bud cell cortex to talk about Kar9-dependent positioning here.In conclusion, this section should be deleted from a revised manuscript to maintain the focus on the findings linking cMTs and Spc72. Also note that the authors could not answer here a number of essential queries from the original review on technical grounds.– Discussion paragraph two. Please revise to simply mention synthetic lethality between kar9 and dyn1 mutants, with dyn1 alone showing a severe growth defect at best. Whether this is due to the demands of aligning the spindle during anaphase (as opposed to S. cerevisiae in which spindle alignment is achieved prior to anaphase) remains highly speculative. The severe growth defect due to dyn1∆ in O polymorpha is consistent with weak SPOC-dependent delay. As stated above, the authors report excess of fragmented nuclei and re-budding in dyn1∆ cells, supporting this idea.

According to the reviewer’s suggestion, we have deleted the former Figure 1—figure supplement 6 that showed data on Kar9 and Dynein pathways as well as SPOC from the Result section. The passage on those in the Introduction has been simplified, and the presence of probable orthologs of Kar9 and Dyn1 is now discussed in the Discussion; “Currently molecular mechanism(s) that regulate spindle orientation is unknown. However, although the timing of spindle orientation relative to cell cycle progression appears to be different from that of other yeasts, two redundant molecular mechanisms of spindle orientation, one requiring dynein and the other Kar9, may be conserved in *O. polymorpha*, because putative orthologs of *KAR9* and dynein were identified in *O. polymorpha* genome sequences (Li et al., 1993; Miller and Rose, 1998; Maekawa and Kaneko, 2014; Nordberg et al., 2014).” The text was also edited; “…which may largely rely on an immediate correction of the orientation of the spindle and on SPOC activity.”

3) The definition of metaphase spindle is not kept consistent throughout. At one point, metaphase is defined by SPB distance < 2 µm but earlier in the text, it was defined as < 1 µm. Please amend accordingly.

We thank the reviewer to point out this inconsistency. It is now amended to be 2 µm.

4) The progressive and gradual loss of Spc72 from SPBs at early stages of the cell cycle is inconsistent with Cdc5 acting as the sole switch in Spc72 recruitment (with reversal triggered by mitotic exit). The authors should make this point clearer when discussing Spc72 detection in G1 cells in Figure 4. Please revise to bring the narration in line with the data.

According to the reviewer’s comment, Spc72 signal in G1 was narrated in the text; “This difference of timing may explain the relatively high and variable intensity of Spc72-GFP at SPB in G1 cells (Figure 4).”

5) In the phosphorylation analysis, the change of tags on Spc72 should be explained in full and justified.

The tag on Spc72 was changed to flag tag for higher efficiency in immunoprecipitation and having less unspecific signals in immunoblotting. These reasons are now described in the figure legend of Figure 6—figure supplement 3; “The flag tag was used for the efficiency of immunoprecipitation and lower unspecific signals in immunoblotting.”

6) The rational for testing Spc72 upon starvation could be made clearer – mating is triggered by starvation in O polymorpha.

We thank the reviewer to point this out. We have added a passage which explains the rational for testing Spc72 in starved cells; “cMT play important roles in yeast mating and karyogamy, which are initiated in G1. Because mating is triggered by nutrient starvation in *O. olymorpha*, we examined cMTs and Spc72 in nutrient starved cells.”

7) Discussion paragraph three, on Nud1 being a possible Cdc5 target – please clarify if you meant that Nud1 contains a consensus site for binding the Cdc5 polo box (rather than a polo box consensus sequence?).

We thank the reviewer to point this out. It meant a polo box binding motif. The text has been modified to state clearer; “Nud1, which has one site matching the consensus sequences of polo box binding site”

8) The authors should discuss why Spc72 overexpression is sufficient to force recruitment to SPBs in their model (data in Figure 5).

According to the reviewer’s comment, discussion on how Spc72 was recruitmented to SPB when overexpressed is now added to the Discussion.

9) The Introduction is incorrect regarding Spc72 and cMTs in S. cerevisiae. Simply put, cMTs at the new SPB are formed after a 1 µm long spindle has formed (certainly not G1/S!, Shaw et al., 1997). Please revise to state correctly that the new SPB acquires Spc72 and cMTs after onset of spindle assembly (also see Juanes et al., 2013).

According to the reviewer’s comment, the passage in the Introduction is revised; “New SPB acquires Spc72 and cMTs after the formation of a 1 µm long spindle”